# *SparsyFed*: SPARSE ADAPTIVE FEDERATED TRAINING

**Adriano Guastella**[*,1]          **Lorenzo Sani**[*,2,3]          **Alex Iacob**[2,3]

**Alessio Mora**[1]          **Paolo Bellavista**[1]          **Nicholas D. Lane**[2,3]

## ABSTRACT

Sparse training is often adopted in cross-device federated learning (FL) environments where constrained devices collaboratively train a machine learning model on private data by exchanging pseudo-gradients across heterogeneous networks. Although sparse training methods can reduce communication overhead and computational burden in FL, they are often not used in practice for the following key reasons: (1) data heterogeneity makes it harder for clients to reach consensus on sparse models compared to dense ones, requiring longer training; (2) methods for obtaining sparse masks lack adaptivity to accommodate very heterogeneous data distributions, crucial in cross-device FL; and (3) additional hyperparameters are required, which are notably challenging to tune in FL. This paper presents *SparsyFed*, a practical federated sparse training method that critically addresses the problems above. Previous works have only solved one or two of these challenges at the expense of introducing new trade-offs, such as clients' consensus on masks versus sparsity pattern adaptivity. We show that *SparsyFed* simultaneously (1) can produce 95% sparse models, with negligible degradation in accuracy, while only needing a single hyperparameter, (2) achieves a per-round weight regrowth 200 times smaller than previous methods, and (3) allows the sparse masks to adapt to highly heterogeneous data distributions and outperform all baselines under such conditions.

## 1 INTRODUCTION

Federated Learning (McMahan et al., 2017) has become a standard technique for distributed training on private data (Yang et al., 2018; Ramaswamy et al., 2019; Pati et al., 2022; Wang et al., 2023; Huba et al., 2022; Bonawitz et al., 2019), particularly on edge devices. Given its application to constrained hardware, mitigating communication and computational overheads—significant in standard FL infrastructures (Kairouz et al., 2021; Bellavista et al., 2021)—remains a key field focus. Practical cross-device FL methods typically assume stateless clients with imbalanced, heterogeneous datasets and constrained, diverse hardware (Wang et al., 2021). Restricted client hardware and low communication bandwidth significantly increase training time compared to centralized methods, elongating hyperparameter tuning (Khodak et al., 2021). Additionally, unknown data distributions and dynamic client availability demand robust optimization methods that can handle these variations. When device availability is constrained, the federated orchestrator may struggle to sample a representative client subset (Eichner et al., 2019; Cho et al., 2020; Li et al., 2020b), inducing trade-offs between sampling ratio and efficiency (Charles et al., 2021b).

Sparse training methods improve computational and communication efficiency by reducing (a) memory footprint and FLOPs during training (Raihan & Aamodt, 2020), and (b) the communication costs (Bibikar et al., 2022). However, applying these methods in cross-device FL is challenging due to client availability and data heterogeneity, which can disrupt the binary mask structure across clients (Qiu et al., 2022). Such inconsistencies hinder consensus on the binary mask, lowering global model performance (Babakniya et al., 2023). Recent approaches address these issues using fixed sparse masks (Huang et al., 2022; Qiu et al., 2022) or dynamic methods involving mask

---
[*] Equal Contribution, correspondence to adriano.guastella2@unibo.it [1] Dipartimento di Informatica - Scienza e Ingegneria, Università di Bologna [2] Department of Computer Science and Technology, University of Cambridge [3] Flower Labs, UK

warmup and refreshing (Babakniya et al., 2023). However, fixed masks reduce the adaptability to unseen distributions, while dynamic methods require careful tuning of additional hyperparameters, like warmup duration and refresh interval. Fixed-mask methods also limit model plasticity—the ability to rewire and adapt to diverse distributions (Lyle et al., 2022; 2023). For example, it is known that in a multi-task or continual learning setting, neural networks can be iteratively pruned to build task-specific sub-networks (Mallya & Lazebnik, 2018), a lost ability when adopting a fixed-mask method. This lack of adaptability makes fixed-mask approaches unsuitable for cross-device FL, where unseen distributions frequently arise. Thus, we argue that sparse training methods for cross-device FL should (a) adopt dynamic masking and (b) remain agnostic to optimization and selection methods while minimizing hyperparameter complexity.

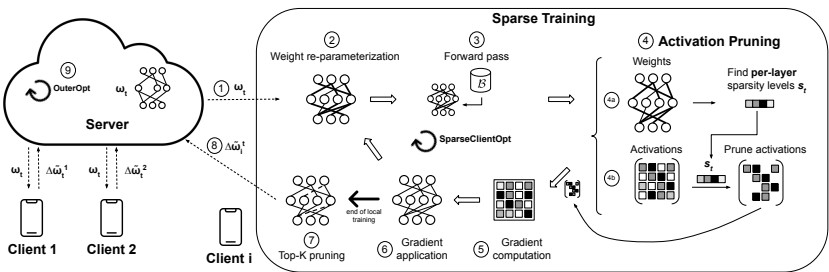

Figure 1: *SparsyFed* pipeline. (1) Server broadcasts the global model $\omega_t$. (2) Client $i$ reparameterizes local weights. (3) Executes a forward pass on batch $\mathcal{B}$. (4a) Computes layer-wise sparsity $s_t$. (4b) Prunes activations using $s_t$ and stores them. (5) Computes grads. (6) Applies grads. (7) Computes model updates and applies `Top-K` pruning. (8) Sends sparse updates $\Delta\tilde{\omega}_i^t$ back to the server. (9) Apply server optimizer to obtain the global model. Steps (2-6) repeat until convergence.

Our work addresses these challenges by introducing *SparsyFed*, a sparse training method for training global models in cross-device federated learning. During local training, as illustrated in Fig. 1, *SparsyFed* uses an easy-to-tune approach to prune (1) activations with an adaptive layer-wise method and (2) model weights before communication using the target sparsity. These strategies can reduce FLOPs and memory usage, with suitable hardware support (Raihan & Aamodt, 2020), and the communication costs by transmitting sparse updates. Our layer-wise approach prunes based on each layer's parameter proportion, unlike previous methods (Babakniya et al., 2023; Qiu et al., 2022), which applied fixed global sparsity across layers. This approach removes more capacity from dense layers while preserving parameter-efficient ones. *SparsyFed* also prunes models at the end of each round, allowing the complete flexibility of the local optimization. *SparsyFed* provides robust initialization for crucial layers, such as embeddings, without excluding layers to preserve performance as done in past works (Qiu et al., 2022; Raihan & Aamodt, 2020). Additionally, *SparsyFed* employs a sparsity-inducing weight re-parameterization (Schwarz et al., 2021) based on a single parameter, enhancing the model's resilience to sparsity and improving adaptivity, which enables changing the global mask for new clients with diverse data distributions. *SparsyFed* is agnostic to the choice of the outer optimizer, treating sparse model updates as pseudo-gradients. It is also compatible with biased client selection policies, allowing training masks to adapt to the training data utilized. Our work's contributions are:

1. We introduce *SparsyFed*, a method which accelerates on-device training in FL. *SparsyFed* achieves high sparsity (up to $95\%$) without sacrificing accuracy through a novel approach combining hyperparameter-free activation pruning and weight re-parameterization.
2. We compare *SparsyFed* against the latest state-of-the-art techniques, demonstrating superior accuracy over sparse training baselines (including those using fixed sparse masks) and, in some cases, surpassing non-pruned baselines at extreme sparsity levels.
3. *SparsyFed* quickly achieves consensus on client sparse model masks, enabling faster global convergence and significantly reducing downlink communication costs (up to $19.29\times$).
4. We evaluate *SparsyFed* with ablation studies on typical cross-device FL datasets, including CIFAR-10/100 (Krizhevsky, 2012) and Speech Commands (Warden, 2018), under various data heterogeneity conditions.

We provide the developed code publicly available in this repository to facilitate result reproducibility and for the community of researchers in the field.

## 2 BACKGROUND

In the following, we describe typical sparse training and weight re-parametrization techniques, which are key components of our work, and discuss their relevance to cross-device FL.

### 2.1 CROSS-DEVICE FEDERATED LEARNING

Cross-device FL (Kairouz et al., 2021) involves the distributed training of a machine learning (ML) model across a population of edge devices exchanging model updates with a central server through heterogeneous networks (McMahan et al., 2017; Li et al., 2020a). Clients in these settings usually possess minimal data samples and very heterogeneous data distributions Kairouz et al. (2021). Given the constraints of edge devices (e.g., limited processing power and battery life) and the heterogeneity of networks (e.g., diverse bandwidth), computational- and communication-efficient training methods are crucial for practical FL (Bonawitz et al., 2019). The research community has proposed means to optimize FL for communication efficiency (Sattler et al., 2019; Jiang et al., 2023) and computational efficiency (Horvath et al., 2021; Niu et al., 2022; Mei et al., 2022). Our approach aims to optimize both by leveraging sparse training.

### 2.2 SPARSE TRAINING

In centralized settings, sparse training tries to learn a sparse model during training to achieve model compression, lower computational demands during training, or faster inference. A typical sparse training pipeline tends to start with a random sparse network and follow a cycle of regular training, pruning, and regrowth (Mocanu et al., 2018; Mostafa & Wang, 2019; Dettmers & Zettlemoyer, 2019; Liu et al., 2020), acting only on model parameters. A more advanced technique relevant to our work, Sparse Weight Activation Training (SWAT) (Raihan & Aamodt, 2020), tailors the forward and backward passes, acting on both activations and model parameters, to induce a sparse weight topology and reduce the computational burden. In each forward iteration, SWAT selects the `Top-K` weights based on magnitude, using only these as the *active* weights. Only the highly activated neurons associated with `Top-K` pruned activations are considered for backpropagation during the backward pass. Notably, full gradients are still applied in the backward pass, allowing updates to both *active* and *inactive* weights. This mechanism enables the dynamic exploration of different network topologies throughout the training process.

### 2.3 WEIGHT RE-PARAMETRIZATION

Weight re-parametrization (Salimans & Kingma, 2016; Li et al., 2019; Gunasekar et al., 2017; Miyato et al., 2018; Vaskevicius et al., 2019; Kusupati et al., 2020; Schwarz et al., 2021; Zhao et al., 2022) in neural network training involves restructuring how weights are represented to improve training dynamics, optimize convergence, or introduce specific properties such as sparsity, without changing the network's architecture. Particularly relevant to our work, Schwarz et al. (2021) propose a sparsity-inducing weight re-parametrization that aims to shift the weight distribution towards higher density near zero, aiding in pruning low-magnitude weights. This simplification emerges by raising the model parameters to the power of $\beta > 1$ during the forward pass while preserving their sign. The re-parametrized weight vector component $w$ is computed as $w = \text{sign}(v) \cdot |v|^{\beta-1}$, where $v$ represents the original weight vector component, and $\beta$ is a scalar value. Due to the chain rule, small-valued parameters receive smaller gradient updates, while large-valued parameters receive more significant updates, reinforcing a "rich get richer" dynamic.

## 3 SPARSE ADAPTIVE FEDERATED TRAINING

In the following, we present our *SparsyFed* method for sparse training in cross-device FL settings. Our method introduces a novel approach based on activation pruning and weight parametrization, applicable to any cross-device FL setting for obtaining a sparse global model. *SparsyFed* reduces the computational and communication overhead of highly heterogeneous FL environments, adapting the sparse mask of the global model. The procedure is outlined in Algorithms 1 and 2.

---

**Algorithm 1** Sparse federated training pipeline of *SparsyFed*.

---

**Require:** $\omega_0$: initial model's weights, $\beta$: weight re-parametrization exponent, $\hat{s}$: target sparsity
**Require:** $T$: number of federated rounds, $E$: number of client local epochs per round
**Require:** $P$: clients population, $\eta_t = \eta(t)$: learning rate scheduler as function of round $t$
**Require:** $\{\mathcal{D}_i\}_{i \in P}$: clients' datasets, $B$: local batch size
 1: **procedure** *SparsyFed*
 2:   **for** $t = 0, \ldots, T - 1$ **do**
 3:    Server samples a subset of clients $S_t \subseteq P$
 4:    **for** each client $i \in S_t$ **in parallel do**
 5:     $\omega_{i,0} \leftarrow \omega_t$
 6:     **for** $k = 0, \ldots, E - 1$ **do**
 7:      $\omega_{i,k+1} \leftarrow \texttt{SparseClientOpt}(\omega_{i,k}, \mathcal{D}_i, B, \beta, \eta_t)$     $\triangleright$ See Alg.2
 8:     $\Delta\omega_i^t \leftarrow \omega_{i,E} - \omega_t$       $\triangleright$ Compute pseudo-gradient
 9:     $\Delta\tilde{\omega}_i^t \leftarrow \texttt{Top-K}(\Delta\omega_i^t, \hat{s})$ $\triangleright$ Prune $\Delta\omega_i^t$ w/ global unstructured $\texttt{Top-K}$ using target $\hat{s}$
10:    $\omega_{t+1} \leftarrow \texttt{OuterOPT}(\omega_t, \{\Delta\tilde{\omega}_i^t\}_{i \in S_t})$    $\triangleright$ Server optimization, e.g., Reddi et al. (2021)
11:   **return** $\omega_T$

---

**Assumptions on the FL setting.** As in any cross-device FL setting, the training is orchestrated by a parameter server (McMahan et al., 2017) that is in charge of initializing the global model, sampling a subset of clients every federated round, aggregating the pseudo-gradients after clients have trained on their local datasets. By following practical considerations (as extensively discussed in Bonawitz et al. (2019); Wang et al. (2021)), *SparsyFed* does not require any particular assumption on the client selection policy, nor on the server optimizer, nor the client optimizer. Thus, our algorithm's design allows it to benefit from any present or future federated optimizer practitioners, and researchers may develop without losing its properties. In particular, *SparsyFed* only requires the addition of one hyperparameter compared to standard dense training, whose sensitivity is discussed in Appendix E.2, making it most suitable for cross-device FL where the hyperparameter optimization (HPO) is challenging (Khodak et al., 2021). We also present a hyperparameter-free alternative in Appendix E.2.2. After initialization, the parameter server iteratively samples clients, broadcasts the latest version of the global model, collects the pseudo-gradients from clients, and aggregates the updates for obtaining the new global model, as described in Algorithm 1. As such, we assume agnosticism w.r.t. the server optimizer (line 10 in Algorithm 1) to allow practitioners to use their preferred one, e.g., $\texttt{ServerOpt}$ in Reddi et al. (2021).

---

**Algorithm 2** Sparse Client Optimization of *SparsyFed*

---

**Require:** $\omega_0$: initial model's weights, $\beta$: weight re-parametrization exponent, $\eta$: learning rate
**Require:** $\mathcal{D} = \{\mathbf{x}_i, y_i\}_{i=1,\ldots,N}$: dataset composed of $N$ samples with inputs $\mathbf{x}_i$ and outputs $y_i$
**Require:** $\mathcal{B}$: batch of data samples, $B$: batch size, $T = \lceil \frac{N}{B} \rceil$: number of batches, $F$: cost function
**Require:** $s(\theta_{t,l})$: function computing the sparsity of the layer $\theta_{t,l}$ at time $t$
**Require:** $size(\theta)$: computes the number of layers in the model $\theta$
**Require:** $\texttt{GetLayer}(a_t, l)$: function extracts the weights and activations for the current layer.
**Require:** $\texttt{SetLayer}(\tilde{a}_{t,l}, l)$: function updates the sparse weights and activations in the model.
 1: **procedure** SPARSECLIENTOPT$(\omega_0, \mathcal{D}, B, \beta, \eta)$
 2:   **for** each step $t = 0, \ldots, T - 1$ **do**
 3:    $\mathcal{B}_t \leftarrow \texttt{GetNextMiniBatch}(\mathcal{D}, B, t)$
 4:    $\theta_t \leftarrow \text{sign}(\omega_t) \cdot |\omega_t|^\beta$   $\triangleright$ Point-wise weights re-parametrization (Schwarz et al., 2021)
 5:    $a_t \leftarrow \texttt{Forward}(\theta_t, \mathcal{B}_t)$      $\triangleright$ Forward pass to compute activations
 6:    **for** each layer $l = 0, \ldots, size(\theta_t) - 1$ **do**
 7:     $\theta_{t,l}, a_{t,l} \leftarrow \texttt{GetLayer}(\theta_t, l), \texttt{GetLayer}(a_t, l)$
 8:     $s_{t,l} = s(\theta_{t,l})$
 9:     $\tilde{a}_{t,l} \leftarrow \texttt{Top-K}(a_{t,l}, s_{t,l})$    $\triangleright$ Prune $a_{t,l}$ w/ unstructured $\texttt{Top-K}$ using target $s_{t,l}$
10:     $\tilde{a}_t \leftarrow \texttt{SetLayer}(\tilde{a}_{t,l}, l)$
11:    $g_t \leftarrow \nabla F(\theta_{t,l}, \tilde{a}_t, \mathcal{B}_t)$     $\triangleright$ Compute gradients using sparse activations
12:    $\omega_{t+1} = \texttt{ClientOpt}(\omega_t, g_t, \eta, t)$   $\triangleright$ Apply client optimizer, e.g., Reddi et al. (2021)
13:   **return** $\omega_T$

---

**Sparsity-Inducing Weights Re-parametrization.** Before the local forward pass at step $t$, we apply a sparsity-inducing re-parametrization to the local model weights $\omega_t$, producing $\theta_t$ (line 4, Algorithm 2). As discussed in Section 2.3, re-parametrization techniques have been widely studied for various purposes. In our case, we aim to enhance pruning effectiveness and efficiency by inducing sparsity directly through optimization and adapting to input data. These outcomes are particularly beneficial in cross-device FL settings, where datasets are highly imbalanced, and some clients may only have a few samples (Section 2.1). From the available sparsity-inducing methods, we adopt Powerpropagation (Schwarz et al., 2021) due to its simplicity (introducing only one hyperparameter) and its ability to preserve the neural network's functional relationships during training. Furthermore, it avoids introducing non-uniform biases across layers, which improves its compatibility with pruning. Every parameter weight $w \in \omega$ is transformed into $v = \text{sign}(w) \cdot |w|^\beta$, where $\text{sign}(w)$ is the sign of $w$, $|w|$ is its L1 norm, and $\beta$ is the Powerpropagation parameter. This re-parametrization enhances global training by promoting client consensus since the dynamics introduced naturally guide training toward the subset of non-zero weights in the model. Weights transitioning from zero to non-zero during training typically have smaller magnitudes, limiting their impact on updates. This ensures that clients focus on a shared subset of weights, facilitating aggregation.

**Pruning Activations During Local Training.** The local training procedure of *SparsyFed* is outlined in Algorithm 2 and relies on two main pillars. First, it ensures that the edge devices benefit from the model's sparsity by reducing memory footprint and FLOPs (depending on the underlying implementation, see Appendix H). Second, it guarantees the retention of as much information as possible during the training. We follow Raihan & Aamodt (2020) in three aspects. First, we use dense activation vectors ($a_t$, line 5) to retain all learned information during the forward pass. Second, we prune activations before the backward pass by aligning their sparsity with the weight vectors (lines 7–10). Specifically, activations are pruned layer-wise using the `Top-K` method with a target per-layer sparsity level ($s_{t,l}$, line 8), determined by the corresponding per-layer weight sparsity (line 9). Such pruning involves retrieving the weights and activations per layer (line 7) and updating the pruned activations before computing gradients. The weight parametrization preserves high sparsity throughout training (Fig. 12, in the appendix), ensuring consistent patterns between weights and activations and seamless integration into the model's sparse structure. Third, we keep gradient vectors ($g_t$, line 11) dense to avoid losing crucial information during updates. Pruning activations before the backward pass reduces computational cost while maintaining dense gradients for robust updates. Notably, initial model weights remain unpruned to allow meaningful training initialization, as clients train a dense model in the first round. With these principles, the local training procedure is designed to be optimizer-agnostic, e.g., compatible with `ClientOpt` (Reddi et al., 2021).

**Pruning model parameters before communication.** The data-driven and hyperparameter-less pruning procedure described above requires a further step to ensure compliance with the communication requirements. Clients receive a model parameters target sparsity, $\hat{s}$, from the server, which must be met before communicating the pseudo-gradient updates. Thus, the client applies a global unstructured pruning step based on `Top-K` using the target value $\hat{s}$ for the output sparsity, guaranteeing to save communication costs using a single parameter. Notably, this allows for non-uniform sparsity across layers, which has been proved to help maintain performance (Kusupati et al., 2020).

## 4 EXPERIMENTAL DESIGN

**Datasets and tasks.** We selected three datasets to assess *SparsyFed*'s performance: CIFAR-10/100 (Krizhevsky, 2012), and Speech Commands (Warden, 2018). CIFAR-10 and CIFAR-100 datasets contain $32 \times 32$ color images of 10 and 100 classes, respectively. The Speech Commands dataset includes audio samples of 35 predefined spoken words and is used for a speech recognition task. Since these datasets' samples can be easily distributed across a pre-defined number of clients, they are common for simulating the heterogeneous data distributions of federated learning settings. On all datasets, we trained models for multi-label classification tasks.

**Data partitioning and sampling.** The datasets above are distributed among 100 clients and partitioned using the method in Hsu et al. (2019), simulating various degrees of data heterogeneity. The distribution of labels across clients is controlled via a concentration parameter $\alpha$ that rules a Latent Dirichlet Allocation (LDA), where a low $\alpha$ value translates to non-IID distribution and a high value to the IID distribution of labels. Specifically, we refer to data distributions as IID for $\alpha = 10^3$ and

non-IID for $\alpha = 1.0$ and $\alpha = 0.1$. To ensure reproducibility, we fixed the seed to 1337 for the LDA partitioning process. The federated orchestrator randomly sampled 10 clients out of the 100 clients in the population every round.

**Model and training implementation.** We employed a ResNet-18 (He et al., 2016) backbone for all experiments, adapting the classification layer to each specific task depending on the number of classes. ResNet-18 was chosen for its size, its popularity in the area, and the scalability of the ResNet family. While our training pipeline is implemented with `PyTorch` (Paszke et al., 2019), we designed custom layers and functions for some of *SparsyFed*'s components, such as layer-wise activation pruning and weight parametrization. We used `Flower` (Beutel et al., 2022) to simulate the federated learning setting. All models were trained from scratch without relying on any pre-trained weights.

**Sparsity ratios.** In our experiments, we targeted different values for the sparsity ratio in the set $\{0.9, 0.95, 0.99, 0.995, 0.999\}$. We chose 0.9 as the minimum value because it has been shown to bring effective gains (Frankle & Carbin, 2019) for both memory footprints and FLOPs (Appendix H). Our investigation spans to the extreme value of 0.999 to fairly present the downsides of these sparsity ratios. We applied the same target sparsity for all devices in the federation. For completeness, we show in Appendix E.6 how *SparsyFed* performs when adopting heterogeneous sparsity targets among devices in the federation.

**Communication costs.** Measuring the communication costs of *SparsyFed* is crucial for understanding its practical implications on cross-device FL settings. To make our analysis agnostic to any compression technique implementation for sparse unstructured models, we report the costs as the number of non-zero parameters effectively exchanged during the FL. Therefore, the communication cost is derived from the effective sparsity of the transmitted model, considering both the downlink (server-to-clients) and uplink (clients-to-server) communication steps. For clarity, we calculate the communication cost as if only one client participated, assuming all communication occurs in parallel. This ensures that our measurements reflect the total communication load without temporal delays between clients and avoids any bias introduced by the varying sampling proportion of clients in each round.

**Reproducibility.** Three different seeds were used for client sampling (5378, 9421, and 2035), while other stochastic processes were seeded with 1337 for reproducibility purposes.

## 5 EVALUATION

This section discusses the evaluation of *SparsyFed* for adaptive sparse training in FL cross-device. The experimental results shown here aim to answer the following research questions.

1. Can *SparsyFed* mitigate the expected accuracy degradation at high and very high sparsity levels? We compare against the baselines, i.e., `Top-K`, ZeroFL (Qiu et al., 2022), and FLASH (Babakniya et al., 2023), for both the accuracy and the communication costs (Sections 5.1 and 5.2, respectively).
2. How does our adaptive sparsity pattern interact with the heterogeneous data distributions compared to other sparse training methods? Similarly to Babakniya et al. (2023), we analyze the consensus across clients on the sparse pattern (Section 5.3).
3. How do the main components of *SparsyFed* contribute to maintaining the accuracy at different sparsity levels? We ablate the activation pruning step (Section 5.5) and vary the weight re-parameterization (Section 5.4) to answer this question.

To ensure a fair and comprehensive comparison, we reimplemented state-of-the-art methods in our experiments, including ZeroFL, FLASH, and `Top-K` pruning. A detailed description of these baseline methodologies and their comparisons against *SparsyFed* is presented in Appendix G.

### 5.1 ACCURACY DEGRADATION

The analysis discussed in this paragraph demonstrates *SparsyFed*'s resilience to the performance degradation typically associated with pruning, as *SparsyFed* achieves competitive results even at high sparsity levels. Compared to other competitive methods under the same sparsity constraints, Table 1 shows that *SparsyFed* exhibits the lowest accuracy drop across all settings and target sparsity

| Dataset | Sparsity | $\alpha = 1.0$ | | | | $\alpha = 0.1$ | | | |
|---|---|---|---|---|---|---|---|---|---|
| | | ResNet-18 | ZeroFL | FLASH | *SparsyFed* | ResNet-18 | ZeroFL | FLASH | *SparsyFed* |
| CIFAR-10 | dense | $83.70 \pm 1.70$ | - | - | - | $73.81 \pm 4.84$ | - | - | - |
| | 0.9 | $80.56 \pm 1.90$ | $76.16 \pm 1.30$ | $81.15 \pm 1.03$ | **82.13 ± 1.53** | $69.79 \pm 3.78$ | $67.40 \pm 4.11$ | $71.87 \pm 2.63$ | **75.00 ± 2.78** |
| | 0.95 | $74.71 \pm 3.29$ | $75.53 \pm 2.27$ | $79.36 \pm 1.03$ | **82.60 ± 1.58** | $60.00 \pm 4.66$ | $61.55 \pm 4.18$ | $72.08 \pm 2.09$ | **75.95 ± 3.39** |
| | 0.99 | $66.27 \pm 5.08$ | $70.71 \pm 0.15$ | $73.45 \pm 1.37$ | **77.71 ± 1.69** | $43.96 \pm 11.99$ | $51.71 \pm 3.54$ | $56.91 \pm 3.55$ | **63.69 ± 3.90** |
| | 0.995 | $63.82 \pm 2.41$ | $56.02 \pm 3.95$ | $69.15 \pm 1.60$ | **70.01 ± 0.43** | $19.02 \pm 10.77$ | $41.33 \pm 3.64$ | $52.15 \pm 3.87$ | **56.79 ± 3.97** |
| | 0.999 | $31.79 \pm 19.10$ | $17.66 \pm 8.34$ | $36.07 \pm 7.49$ | **51.39 ± 3.19** | $11.50 \pm 4.49$ | $18.76 \pm 4.28$ | $29.31 \pm 6.75$ | **43.68 ± 7.61** |
| CIFAR-100 | dense | $52.29 \pm 1.14$ | - | - | - | $48.34 \pm 2.71$ | - | - | - |
| | 0.9 | $46.57 \pm 1.71$ | $40.70 \pm 4.72$ | $51.99 \pm 0.21$ | **53.08 ± 0.90** | $41.96 \pm 2.16$ | $31.92 \pm 7.65$ | $45.59 \pm 0.75$ | **48.37 ± 1.73** |
| | 0.95 | $28.07 \pm 23.27$ | $38.82 \pm 1.75$ | $47.19 \pm 1.88$ | **52.81 ± 1.72** | $11.48 \pm 17.51$ | $34.21 \pm 7.65$ | $44.31 \pm 2.14$ | **48.27 ± 2.70** |
| | 0.99 | $19.65 \pm 16.30$ | $18.97 \pm 2.08$ | $42.76 \pm 4.08$ | **46.64 ± 1.59** | $0.14 \pm 0.72$ | $13.07 \pm 2.26$ | $34.75 \pm 3.38$ | **41.03 ± 2.14** |
| | 0.995 | $9.51 \pm 14.81$ | $6.01 \pm 4.74$ | $36.43 \pm 4.97$ | **42.21 ± 1.03** | $0.14 \pm 0.72$ | $7.04 \pm 5.25$ | $26.44 \pm 17.35$ | **35.72 ± 2.01** |
| | 0.999 | $3.81 \pm 2.18$ | $1.96 \pm 0.66$ | $5.80 \pm 2.86$ | **15.96 ± 0.64** | $0.14 \pm 0.72$ | $1.66 \pm 0.97$ | $3.56 \pm 2.07$ | **13.84 ± 3.69** |
| Speech Commands | dense | $91.49 \pm 0.94$ | - | - | - | $80.15 \pm 2.69$ | - | - | - |
| | 0.9 | $84.28 \pm 0.88$ | $87.79 \pm 1.40$ | $88.68 \pm 1.72$ | **92.32 ± 1.59** | $65.44 \pm 0.97$ | $70.35 \pm 2.65$ | $77.15 \pm 0.77$ | **79.67 ± 2.78** |
| | 0.95 | $78.58 \pm 0.44$ | $84.29 \pm 1.50$ | $84.89 \pm 0.49$ | **89.14 ± 1.15** | $57.39 \pm 1.04$ | $65.90 \pm 1.88$ | $71.28 \pm 1.75$ | **75.46 ± 2.24** |
| | 0.99 | $65.01 \pm 0.84$ | $57.79 \pm 0.82$ | $69.22 \pm 1.59$ | **75.82 ± 3.72** | $50.42 \pm 6.26$ | $41.42 \pm 1.60$ | $53.55 \pm 2.00$ | **56.69 ± 4.56** |
| | 0.995 | $56.73 \pm 1.00$ | $37.16 \pm 2.71$ | $58.23 \pm 1.84$ | **68.02 ± 3.14** | $34.20 \pm 1.43$ | $22.61 \pm 3.45$ | $43.16 \pm 3.47$ | **48.30 ± 5.39** |
| | 0.999 | $21.56 \pm 12.79$ | $10.10 \pm 4.01$ | $17.70 \pm 2.58$ | **47.43 ± 1.66** | $19.25 \pm 6.01$ | $8.85 \pm 3.76$ | $17.14 \pm 2.97$ | **29.24 ± 2.34** |

Table 1: Aggregated results for CIFAR-10, CIFAR-100, and Speech Command datasets, with ResNet-18, ZeroFL, FLASH, and *SparsyFed* implementations.

levels. A noticeable drop in performance compared to the dense model was observed only at $99\%$ sparsity. This advantage arises from using weight re-parameterization, which, in some cases, can even enhance the performance of the dense model. Additionally, the minimal performance drop at lower sparsity levels ($90 - 95\%$) allows *SparsyFed* to outperform the dense model in specific scenarios. To stress more our method capabilities, we increased the target sparsity to the point ($99.9\%$) where *SparsyFed* is no longer able to retain sufficient accuracy. Our results show that all the baselines struggle to train effectively under such conditions. The adaptivity of *SparsyFed*'s sparsity patterns promotes consistent performance across clients, even in highly sparse settings, leading to a more synchronized and globally pruned model.

## 5.2 COMMUNICATION COSTS

The promising accuracy achieved at very high sparsity ratios makes *SparsyFed* particularly suitable for cross-device FL settings, where communication costs are a critical concern. As illustrated in Fig. 2 (left), *SparsyFed* significantly outperforms the baselines regarding both communication savings ($19.29\times$ less communication costs compared to the dense model and $1.66\times$ compared to ZeroFL) and preserving required accuracy (consistently above $45\%$). FLASH has comparable communications costs, $0.97\times$ ours, but results in lower accuracy. Notably, *SparsyFed* consistently achieves higher accuracy for a given communication cost than the baselines, as shown in Fig. 2 (left). This advantage arises from *SparsyFed*'s ability to prune weights during client training and maintain a close-to-target sparsity ratio during server aggregation, effectively reducing uplink and downlink communication costs. Importantly, FLASH does not see an increase in model density after aggregation due to its fixed-mask local training regime, which prevents weight regrowth. This characteristic results in stable communication costs. In contrast, ZeroFL experiences a substantial increase in model density after aggregation, leading to a systematic and significant increase in downlink communication costs.

## 5.3 CONSENSUS ON THE SPARSE MASKS ACROSS CLIENTS

Achieving consensus on the sparse masks of the clients' updates after each training round is crucial for achieving high accuracy. This consensus dynamic can be interpreted as the global model stabilizing around the target sparsity level as federated rounds progress. As shown in Fig. 2 (right), *SparsyFed* demonstrates minimal deviation from the target sparsity ($90\%$), whereas ZeroFL and `Top-K` drop below $47\%$ and $83\%$, respectively, during the initial training stage. *SparsyFed*'s consistency allows clients to effectively collaborate in training the same subset of parameters, which reduces the need for excessive pruning after local training and helps retain more helpful information. Such consistency appears despite our clients being allowed to dynamically change their mask during training, unlike fixed-mask approaches. For all methods, local updates maintain consistent sparsity due to post-training pruning; therefore, any increase in the global model's density indicates that local weight regrowth during training alters the local masks. Once these altered updates are ag-

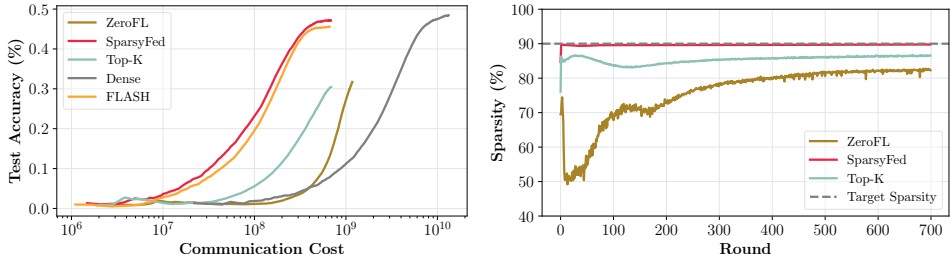

Figure 2: **(left)** The plot on the left compares accuracy versus communication cost for four implementations: ZeroFL, `Top-K`, FLASH, and *SparsyFed*, with the dense model as a reference. The test is conducted on CIFAR-100 partitioned with LDA($\alpha = 0.1$) and $95\%$ sparsity. *SparsyFed* outperforms the baselines, achieving high accuracy and communicating less. **(right)** The plot on the right shows the global model sparsity level, measured on the server after aggregating local updates (CIFAR-100, $\alpha = 0.1$). The density gain reflects mismatches between client updates, causing the aggregated model to regain density, which can degrade performance and increase downlink communication. **Note**: FLASH maintains target sparsity after the first round with a fixed mask.

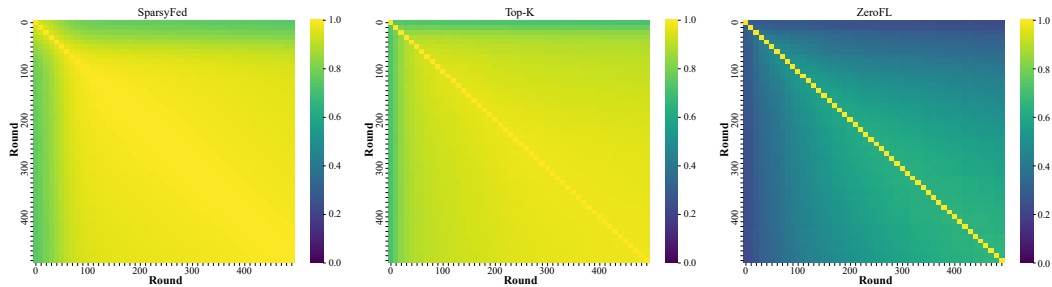

Figure 3: Intersection over Union (IoU) of global model binary masks between training rounds for *SparsyFed*, `Top-K`, and ZeroFL (CIFAR-100, $\alpha = 0.1$, 95% target sparsity). The IoU is calculated between each mask and all other masks across rounds to show changes over time. The x and y axes represent training rounds indices–the diagonal indicates the identity. Higher IoU values (close to 1.0) signify stronger similarity between masks, while lower values indicate significant changes. *SparsyFed* shows consistent mask movement with minimal variation, suggesting strong consensus on weight usage among clients. ZeroFL struggles to find mask consensus, with masks continuing to shift even in later rounds. **Note:** FLASH is absent since the global mask is fixed.

gregated, the result is a denser global model on the server. This misalignment among clients' masks can negatively impact both communication efficiency and accuracy. Global training appears to benefit from this consensus in terms of convergence, as shown in Fig. 2 (left), where both *SparsyFed* and FLASH outperform `Top-K` despite similar overall communication costs.

### 5.3.1 SPARSITY PATTERN DYNAMICS

We analyze the global model's sparse mask dynamics during training to shed light on *SparsyFed*'s ability to maintain a stable sparse pattern driven by two key factors: (1) focused weight utilization and (2) sparse mask adaptivity. First, the weight re-parameterization method targets a critical subset of weights, concentrating information where it is most impactful. This targeted approach improves training efficiency, enhances client collaboration, and reduces communication costs, as shown in Fig. 2. Second, unlike fixed-mask methods like FLASH, *SparsyFed* promotes dynamic mask adaptation, achieving robustness to heterogeneous data distributions throughout the training process. This allows a natural warm-up phase during which the sparsity pattern emerges organically, as shown in Fig. 3 (left). Clients rapidly converge on a stable shared mask, ensuring consistent performance. In contrast, on Fig. 3 (center), `Top-K` never fully settles on a stable mask, resulting in continuous variation across rounds while being faster in its initial rounds. In Fig. 3 (right), ZeroFL,

though more robust initially, struggles to maintain performance as it forces weight regrowth to adjust the mask, leading to instability in later rounds.

We do not consider FLASH in this comparison since it fixes its mask after the first round, eliminating any changes in the sparse mask pattern but reducing mask adaptivity entirely. While this strategy ensures that the model adheres to a fixed sparse structure, it introduces potential drawbacks. Specifically, since the mask is determined based on the data distribution observed in the first round, FLASH becomes highly dependent on the initial client data. This lack of adaptivity contrasts with our method, which allows continuous changes. Thus, it is more effective at handling shifts in data distribution across rounds.

## 5.4 ABLATION ON WEIGHT RE-PARAMETERIZATION

In this ablation study, we evaluated various weight re-parameterization techniques for their ability to sustain a sparsity-driven model while preserving dense-like performance. Since the sparse activations during the backward pass rely on a sparse weight model, the model must maintain high sparsity levels throughout training while minimizing accuracy loss.

We evaluated three approaches: fixed-mask training, spectral re-parameterization (Miyato et al., 2018), and Powerpropagation (Schwarz et al., 2021). We provide more details on the spectral re-parameterization in Appendix D.3. Each technique was applied to a ResNet-18 model trained with sparse activations for the backward pass and `Top-K` unstructured pruning at the end of each training round. This setup enabled direct comparison with a baseline model that lacked any re-parameterization. Among the methods, Powerpropagation proved to be most effective for our use case (Fig. 4), demonstrating superior resilience to preserve the accuracy with minimal degradation. By leveraging a "rich get richer" dynamic, Powerpropagation naturally induces sparsity during training. This enables the model to remain sparse during training, complementing our sparse activation backward pass and improving the model's overall efficiency and effectiveness.

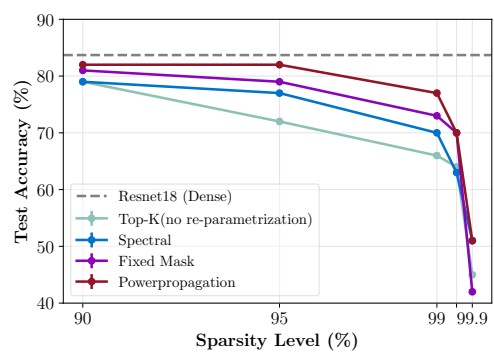

Figure 4: We report the test accuracy of different re-parameterization methods with sparse activations during backpropagation. We deployed a ResNet-18 trained on the CIFAR-10 dataset using LDA($\alpha = 1$). This plot illustrates the methods' performance under different sparsity levels. Powerpropagation exhibited superior robustness to the applied sparsity levels, achieving the best overall performance among these methods.

## 5.5 ABLATION ON ACTIVATION PRUNING

The activation pruning step in *SparsyFed* (lines 6–10 in Algorithm 2) is designed to reduce computational costs of local training, which is particularly sensitive when executing on edge devices. This decision is motivated by prior studies Section 2 and Appendix H. As part of our ablation studies, we analyzed the impact of this pruning step on the accuracy and overall performance of the model. Since pruning activations during the backward pass do not directly affect weight density, we focus explicitly on its influence on test accuracy.

In Table 2, we compare *SparsyFed* with and without activation pruning during the backward pass. Our results show that activation pruning minimally impacts test accuracy at higher density levels. Significant performance degradation occurred only under extreme sparsity, where excessive pruning in specific layers substantially reduced activations, leading to diminished overall performance. Ultimately, the computational speedup achieved through activation pruning validates its inclusion in *SparsyFed*, as it balances efficiency with model accuracy.

| Sparsity | $\alpha = 10^3$ (IID) | | $\alpha = 1.0$ (non-IID) | | $\alpha = 0.1$ (non-IID) | |
|---|---|---|---|---|---|---|
| | no Act. Pr. | Act. Pr. | no Act. Pr. | Act. Pr. | no Act. Pr. | Act. Pr. |
| 0.90 | $83.92 \pm 1.58$ | $\mathbf{84.31 \pm 0.86}$ | $\mathbf{82.27 \pm 2.21}$ | $82.13 \pm 1.23$ | $\mathbf{76.60 \pm 1.54}$ | $75.00 \pm 2.78$ |
| 0.95 | $83.80 \pm 0.90$ | $\mathbf{84.25 \pm 1.38}$ | $81.53 \pm 2.10$ | $\mathbf{82.6 \pm 1.58}$ | $75.29 \pm 2.64$ | $\mathbf{75.95 \pm 3.39}$ |
| 0.99 | $\mathbf{77.54 \pm 1.98}$ | $77.16 \pm 0.85$ | $75.76 \pm 1.78$ | $\mathbf{77.71 \pm 1.69}$ | $\mathbf{63.79 \pm 3.96}$ | $63.69 \pm 3.90$ |
| 0.995 | $\mathbf{74.6 \pm 1.01}$ | $72.71 \pm 0.65$ | $\mathbf{70.89 \pm 2.22}$ | $70.01 \pm 0.43$ | $\mathbf{59.15 \pm 2.49}$ | $56.79 \pm 3.97$ |
| 0.999 | $\mathbf{62.12 \pm 1.74}$ | $55.24 \pm 2.09$ | $\mathbf{62.67 \pm 2.19}$ | $51.39 \pm 3.191$ | $\mathbf{49.43 \pm 2.45}$ | $43.68 \pm 7.61$ |

Table 2: Accuracy comparison between *SparsyFed* with and without the pruning of the activation, on CIFAR-10 with LDA $\alpha = 10^3$, $\alpha = 1.0$, and $\alpha = 0.1$.

## 6 RELATED WORK

**Sparse training in centralized settings.** Methods to enforce sparsity in neural networks can be grouped into two main categories: (1) dense-to-sparse methods that train a dense model and achieves a sparse model after training (Molchanov et al., 2017; Louizos et al., 2017), (2) sparse-to-sparse methods where pruning happens during training (Louizos et al., 2018; Dettmers & Zettlemoyer, 2019; Evci et al., 2020; Jayakumar et al., 2021; Raihan & Aamodt, 2020), thus theoretically reducing computational requirements (Bengio et al., 2015) and speeding up training. Our work introduces a novel method to implement sparse training directly within FL clients, inspired by sparse-to-sparse approaches in centralized settings such as Raihan & Aamodt (2020).

**Pruning model updates in FL.** After-training pruning, where clients first train dense models and then prune updates, is a common approach in FL (Sattler et al., 2019; Wu et al., 2020; Malekijoo et al., 2021). Sparse Ternary Compression (Sattler et al., 2019) combines `Top-K` pruning with ternary quantization to compress client updates, while FedZip (Malekijoo et al., 2021) uses layer-wise pruning, and FedSCR (Wu et al., 2020) employs patterns in client updates for more aggressive compression. However, these methods primarily improve communication efficiency without reducing computational overhead, as they still train dense models. In contrast, our method trains sparse models from the start, resulting in sparse updates and maintaining sparsity even after server aggregation, ensuring efficient upstream and downstream communication.

**Sparse training in FL.** Several studies have explored sparse learning in federated settings (Bibikar et al., 2022; Huang et al., 2022; Jiang et al., 2023; Qiu et al., 2022; Babakniya et al., 2023), but each has limitations. FedDST (Bibikar et al., 2022) applies RigL (Evci et al., 2020) to train sparse models, focusing on heterogeneous data without addressing extreme sparsity levels. FedSpa (Huang et al., 2022) uses a fixed sparse mask throughout training without a clear rationale behind it. PruneFL (Jiang et al., 2023) computes a sparse mask using biased client data and requires full gradient uploads, increasing communication costs. ZeroFL (Qiu et al., 2022) integrates SWAT (Raihan & Aamodt, 2020) for local training but struggles with weight regrowth, requiring pruning after each round, which can cause information loss. FLASH (Babakniya et al., 2023) introduces a fixed mask after a warm-up phase, but it depends heavily on the chosen clients during the first sampling and does not adapt to concept drift. In contrast, our method uses a dynamic sparse mask, offering more flexibility and better performance in highly non-IID FL settings.

## 7 CONCLUSIONS

This work presents *SparsyFed*, an adaptive sparse training method tailored for cross-device federated learning (FL). We show that *SparsyFed* can achieve impressive sparsity levels while minimizing the accuracy drop due to the compression. *SparsyFed* outperforms in accuracy three federated sparse training baselines, `Top-K`, ZeroFL, and FLASH, using adaptive and fixed sparsity for three typical datasets used in cross-device FL. We were able to ensure a limited drop in accuracy at sparsity levels of up to 95%, achieving up to a $19.29\times$ reduction in communication costs compared to dense baselines. The results presented in this work make our proposal particularly suitable for cross-device FL settings, which may require extreme communication cost reductions and the capability to adapt to heterogeneous distributions across federated rounds.

ACKNOWLEDGMENTS

This research was supported by the following entities: The Royal Academy of Engineering via DANTE (a RAEng Chair); the European Research Council, specifically the REDIAL project; Google through a Google Academic Research Award; in addition to both IMEC and the Ministry of Education of Romania (through the Credit and Scholarship Agency); and partly supported by the European Union under the NRRP partnership on "Telecommunications of the Future" (PE00000001 - program "RESTART") and by the National PRIN JOULE project. Furthermore, we thank Javier Fernandez-Marques, Stefanos Laskaridis, Xinchi Qiu, and Francesco Corti for their invaluable feedback and collaborative spirit throughout this research project and the ICLR reviewers for helping improve our paper.

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

## A    IMPLEMENTATION

We implemented all experiments using a ResNet-18 backbone (He et al., 2016), which was adjusted to accommodate the number of classes for each task. The training pipeline was developed using `PyTorch` (Paszke et al., 2019), and specific components of the various methods, such as layer-wise activation pruning and weight re-parameterization, were implemented through custom layers and functions.

To simulate the federated learning environment, we used the `Flower` framework (Beutel et al., 2022). All models were trained from scratch, without pre-trained weights, to ensure a fair evaluation of *SparsyFed*'s performance under different sparsity settings. We only performed fine-tuning of a pre-trained model for the experiments on Visual Transformer (ViT) models.

We utilized a template from CaMLSys Lab to structure the code and ensure easy reproducibility. The complete code for the implementation is available in this repository.

## B    RELATED WORK

**Sparse training in centralized settings.**    Methods to enforce sparsity in neural networks can be grouped into two main categories: (1) dense-to-sparse methods that train a dense model and achieves a sparse model after training (Molchanov et al., 2017; Louizos et al., 2017), (2) sparse-to-sparse methods where pruning happens during training (Louizos et al., 2018; Dettmers & Zettlemoyer, 2019; Evci et al., 2020; Jayakumar et al., 2021; Raihan & Aamodt, 2020), thus theoretically reducing computational requirements (Bengio et al., 2015) and speeding up training. Our work introduces a novel method inspired by sparse-to-sparse approaches in centralized settings such as Raihan & Aamodt (2020) to implement sparse training directly within FL clients.

**Pruning model updates in FL.**    After-training model pruning, where clients regularly train their dense model and then apply pruning to the resulting updates, has been widely explored in the FL literature (Sattler et al., 2019; Wu et al., 2020; Malekijoo et al., 2021). Sattler et al. (2019) introduced Sparse Ternary Compression, which combines `Top-K` pruning with ternary quantization on client updates. Similarly, FedZip (Malekijoo et al., 2021) applies a layer-wise pruning approach, while FedSCR (Wu et al., 2020) leverages patterns in client updates for more aggressive compression. However, these methods primarily focus on enhancing communication efficiency without tackling the issue of reducing computational overhead during training, as they conduct regular training on a dense model before pruning weight updates. Our approach simultaneously addresses both concerns, as training a sparse model naturally yields sparse updates. Furthermore, existing after-training pruning methods typically focus solely on upstream communications or experience performance degradation when applied downstream. In contrast, our method effectively maintains sparsity after server-side aggregation, as clients rapidly converge on a shared sparsity mask, ensuring that server-to-client payloads remain sparse.

**Sparse training in FL.**    Few studies have investigated the benefits of sparse learning in federated settings (Bibikar et al., 2022; Huang et al., 2022; Jiang et al., 2023; Qiu et al., 2022; Babakniya et al., 2023). Specifically, FedDST (Bibikar et al., 2022) used RigL (Evci et al., 2020) to train sparse models on clients but mainly focused on highly heterogeneous data distributions without considering extreme sparsity levels. FedSpa (Huang et al., 2022) employed a randomly initialized sparse mask that remained fixed throughout training, providing no clear justification for this approach. PruneFL (Jiang et al., 2023) computes the sparse mask during the initial round on a designated client using its potentially biased data. Qiu et al. (2022) empirically found that, after local training, the `Top-K` weights differ across clients, particularly at higher sparsity levels, leading to divergent sparse masks. This divergence makes aggregation inefficient and results in downstream dense models. In response, Qiu et al. (2022) proposed ZeroFL, which integrates unstructured SWAT during local client training. However, this alone does not guarantee achieving the desired sparsity level, as SWAT often leads to weight regrowth with each optimizer step. To address this, ZeroFL applies `Top-K` pruning before sending the model back to the server, ensuring the model meets the targeted sparsity. It is essential to highlight that ZeroFL applies the same sparsity level uniformly across all model layers, regardless of their sensitivity. The recent work in Babakniya et al. (2023) introduces FLASH, which employs a fixed binary mask throughout the training process. This mask is established during a warm-up

phase where a subset of clients ($[10, 20]$) train their model for a non-negligible number of epochs ($[10, 20, 40]$), and compute their per-layer sensitivity. The server aggregates the client's sensitivities to determine the mask. This mask's fixed nature means no weight regrowth is allowed, and no further pruning is required after the warm-up phase.

**Cross-device federated learning.** Federated Learning (FL) has emerged as a paradigm shift from traditional machine learning approaches, improving data privacy and moving the computation load to the network's edge. FL enables collaborative model training across decentralized devices while keeping data localized, thereby mitigating risks associated with centralized data aggregation (McMahan et al., 2017). In FL, client devices participate in model training via iterative rounds, aggregating local updates to build a global model. This distributed approach is advantageous for scenarios involving edge devices with limited computational resources and intermittent connectivity (Kairouz et al., 2021). Cross-device Federated Learning (FL) is particularly challenging due to the heterogeneous and resource-constrained nature of client devices, such as smartphones and IoT devices. The variability in hardware capabilities and data distributions across devices necessitates specialized techniques to optimize computation and communication techniques, such as quantization and model pruning, which have shown promise in reducing the amount of data that must be transmitted during each communication round. However, these methods often face challenges in maintaining model performance and achieving consensus on the sparsity patterns among clients.

**Model pruning.** Sattler et al. (2019) propose Sparse Ternary Compression (STC), a lossy compression scheme able to reduce the per-round communication cost of FL iterations significantly. STC first applies `Top-K` pruning after unaltered on-device training and then further compresses the weight updates using a ternary quantization. Other types of work have followed the idea proposed in Rigging the Lottery Ticket (Evci et al., 2020), where an initial sparse mask is fixed at the beginning of the training and remains primarily unchanged throughout. This approach allows clients to train only the initially fixed weights (Jiang et al., 2023; Babakniya et al., 2023).

**Weight parametrization.** Re-parameterization of weights in machine learning refers to the process of altering the re-parameterization of a model's weights to achieve various goals or facilitate specific training strategies (Li et al., 2019; Gunasekar et al., 2017; Zhao et al., 2022; Vaskevicius et al., 2019). This approach can be beneficial for enhancing the model's robustness to the application of sparsity. This work considers a re-parameterization based on Weight Spectral Normalization (Miyato et al., 2018) and Powerpropagation (Schwarz et al., 2021). The former re-parameterizes the weights based on the proportion of each weight relative to the one with the highest magnitude. At the same time, the latter applies an alpha power to the weights, inducing a "rich get richer" mechanism.

$$W_{\text{refactor}} = W \cdot \frac{W}{\sigma(W)}$$

## C  ADDITIONAL EXPLANATIONS ON *SparsyFed* COMPONENTS

**Model preparation for pruning.** The first step is to prepare the model for pruning by applying a re-parameterization to the weights at the layer level, as proposed in Schwarz et al. (2021). This approach leverages the information already present in the model by applying power to a specific value, $\beta$, to the network weights. This re-parameterization aims to induce a sparse representation of the weight of the network.

Following the original approach, each weight $w_i$ is replaced by $w_i^\beta$. This transformation only applies to the neural network weights, leaving other parameters unchanged. Given the re-parameterized loss function $L(\mathbf{w}^\beta)$, the gradient with respect to $\mathbf{w}$ becomes:

$$\nabla_{\mathbf{w}} L(\mathbf{w}^\beta) = \nabla_{\mathbf{w}^\beta} L(\mathbf{w}^\beta) \cdot \beta \mathbf{w}^{\beta-1}$$

Here, $\nabla_{\mathbf{w}^\beta} L(\mathbf{w}^\beta)$ is the gradient concerning the re-parameterized weights. This gradient is scaled element-wise by $\beta \mathbf{w}^{\beta-1}$, which adjusts the step size proportionally to the magnitude of each weight. This update is distinct from simply scaling the gradients in the original re-parameterization, as it

directly modifies the re-parameterized weights, not the original weights. This modified gradient step enables a "rich-get-richer" dynamic where the gap between high and low-magnitude weights increases, creating a natural separation between them. This makes the model relatively insensitive to pruning.

**Weight pruning.** Our first concern was to reduce the model's size during communication rounds to speed up FL training. In this context, a compression mechanism is typically used to reduce the payload size that has to be exchanged. To achieve this, reducing the number of parameters in the network is necessary, inducing a high level of sparsity. We used a `Top-K` pruning method to remove all low-magnitude weights from the network. This pruning operation is implemented at the end of each local training on the client. This decision is based on several observations: (a) since the parametrization affects gradient descent, pruning before training would not be beneficial, (b) the server is not aware of the features of the data on the clients, and initializing the sparsity mask on the server side could cause a significant drop in performance.

We also decided to induce sparsity from the very first round. The clients would only use a small portion of the weights during training, so they must start training with the restricted fraction to maximize their effectiveness. Sparsity is induced from the first round of local training on each device, allowing the local updates to be sparse from the beginning of the federated training.

During the following training rounds, a second quality of Powerpropagation comes into play, drastically reducing the regrowth of new weights during training. This results in minimal shifts of the model's sparsity mask from the global one received from the server. Thus, the sparsity mask of the new model update sent to the server will differ very little from the global one. In other words, using Powerpropagation, we are forcing all clients to converge towards a shared sparsity mask, similar to Evci et al. (2020), without inhibiting the regrowth of new paths. The clients decide the mask in the first round of training based on their local data. This means that the global model will remain highly sparse even after aggregating new local updates from the clients. This result allows for significant compression during the downlink communication from the server to the clients in subsequent training rounds.

**Activation pruning.** To further reduce the computation footprint during the training, we implemented a layer-wise pruning similar to the one proposed in Raihan & Aamodt (2020). The original proposal was to speed up inference and training by reducing the number of computational operations, inducing a fixed level of sparsity on the weights of each layer during both the forward and backward passes. However, this could cause a significant drop in performance with high sparsity levels, as not all layers retain the same level of information. Applying the same sparsity to all layers could negatively impact those that naturally retain more information. In our implementation, we heavily modified the approach, retaining only key concepts. Since we keep the model sparse during almost all training rounds, except for the first, pruning the weights during the forward pass is not necessary.

For the pruning of activations during the backward pass, instead of applying the same level of sparsity to all layers of the model, the global sparsity mask is used to determine the pruning sensitivity of each layer. The level of pruning applied to the activations is directly proportional to the level of the sparsity of the weights in the same layer, allowing layers that retain more information to maintain denser activations while drastically reducing the activations in layers that reach a high level of sparsity.

# D  ADDITIONAL BACKGROUND

## D.1  POWERPROPAGATION

Powerpropagation is a weight re-parameterization technique that induces sparsity in neural networks. Essentially, it causes gradient descent to update the weights in proportion to their magnitude, leading to a "rich get richer" dynamic where small-valued parameters remain largely unaffected by learning. As a result, models trained with Powerpropagation exhibit a weight distribution with significantly higher density at zero, allowing more parameters to be pruned without compromising

performance. Powerpropagation involves raising model parameters to the power of $\beta$ (where $\beta > 1$) in the forward pass while preserving their sign. This re-parameterization can be expressed as

$$w = v \cdot |v|^{\beta-1}$$

where $w$ represents the weights, and $v$ are the re-parameterized parameters. Due to the chain rule, this re-parameterization causes the magnitude of the parameters (raised to $\beta-1$) to appear in the gradient computation. Consequently, small-valued parameters receive smaller gradient updates, while large-valued parameters receive larger updates, thus amplifying the "rich get richer" dynamic. Powerpropagation leads to intrinsically sparse networks, meaning that a significant portion of the weights converge to values near zero during training. This property allows for the pruning (removal) of many weights without significantly compromising model performance. Powerpropagation can also be easily integrated with existing sparsity algorithms to enhance performance further. Studies show the benefits of combining Powerpropagation with popular methods such as Iterative Pruning and TopKAST (Gale et al., 2019; Jayakumar et al., 2021).

### D.2 FIXED MASK SPARSE TRAINING (FLASH)

Training a fixed subset of weights can be a practical approach for sparse model training in Federated Learning (FL), offering several significant benefits. Limiting the number of active (non-zero) weights reduces the computational and memory demands compared to dense models, which is particularly advantageous given the resource constraints often found in client devices within FL. In addition to these resource savings, sparse models enhance communication efficiency between clients and servers. Since only the active weights need to be transmitted, the message size is reduced, leading to faster training and lower bandwidth usage.

Another advantage of this method is its potential to uncover "winning tickets" within neural networks Frankle & Carbin (2019). Research indicates that dense, randomly initialized networks often contain sparse sub-networks, known as "winning tickets", which, when trained independently, can achieve performance similar to the full model. Training a fixed subset of weights promotes this sparsity and encourages the model to learn more efficient representations, potentially revealing these sub-networks.

Despite these benefits, specific challenges arise when using a fixed subset of weights in FL. If the mask is not initialized correctly or fails to adapt during training, the model's performance may be compromised. To mitigate these issues, strategies such as sensitivity-based pruning and selective mask updates are crucial for fully leveraging the advantages of sparse learning in FL Babakniya et al. (2023). The training process for sparse models can be formalized as:

$$W_{\text{sparse}} = M \odot W$$

where $W_{\text{sparse}}$ represents the sparse weights, $M$ is the binary mask (with values 0 or 1), and $\odot$ denotes element-wise multiplication.

### D.3 SPECTRAL NORMALIZATION

Spectral Normalization (SN) is a weight normalization technique that aims to limit the most significant singular value of each weight matrix, thereby controlling the Lipschitz constant of the function represented by the network. This is achieved by constraining the spectral norm of each layer $g : h_{in} \rightarrow h_{out}$. Formally, given a weight matrix $W$, its spectral norm is defined as the most significant singular value $\sigma(W)$, i.e.,

$$\sigma(W) = \max_{h \neq 0} \frac{\|Wh\|_2}{\|h\|_2}$$

SN normalizes the weight matrix $W$ by dividing it by its spectral norm:

$$\bar{W} = \frac{W}{\sigma(W)}$$

This ensures that the Lipschitz constant of each layer remains bounded, promoting stability during training, particularly in scenarios where gradients may explode. Originally proposed as a regularization technique to stabilize discriminator training in Generative Adversarial Networks (GANs) Miyato et al. (2018), SN has since found broader applications, including improvements in generative neural networks Zhao et al. (2018), and has been theoretically linked to enhanced generalization and adversarial robustness Farnia et al. (2018); Sokolic et al. (2017); Cisse et al. (2017).

In the context of federated learning and sparse training, SN can significantly improve the resilience of models to pruning. SN regularizes the model mappings by enforcing the Lipschitz constraint, reducing sensitivity to high sparsity levels. This effect has been observed in prior work on pruning, where SN was used to prune redundant mappings and enforce a spectral-normalized identity prior Lin et al. (2020).

However, by strictly constraining the Lipschitz constant, SN may reduce the model's flexibility during training, potentially affecting convergence in some cases. To address this, following an approach similar to those proposed in other re-parameterization works Vaskevicius et al. (2019), we propose a slightly different method where the spectral norm is modified to enhance the distribution of the weights. Specifically, we redefine the weight update rule as:

$$w = w \cdot \frac{|w|}{\sigma(W)}$$

This approach allows us to maintain most of the model's performance while improving its resilience to pruning.

# E    ADDITIONAL EXPERIMENTAL RESULTS

## E.1    NAIVE POWERPROPAGATION FEDERATED ADAPTATION

In the original Powerpropagation paper, pruning was applied at the end of centralized training after training a dense model. To create a baseline for comparison, we implemented a naive federated version of Powerpropagation that follows this methodology. In this version, pruning is not applied during local training at the client level. Instead, clients train with the full model, sending and receiving full model updates. At the end of the federated training (after the final round), pruning is applied to the global model using the `Top-K` method.

We compare this naive version of federated Powerpropagation to our proposed method, *SparsyFed*, across various sparsity levels and different levels of data heterogeneity, in Table 3. As shown in the results, *SparsyFed* significantly outperforms the naive federated Powerpropagation in terms of performance and stability, even in sparsity.

| Sparsity | $\alpha = 10^3$ (IID) | | $\alpha = 1.0$ (non-IID) | | $\alpha = 0.1$ (non-IID) | |
|---|---|---|---|---|---|---|
| | Naive PP | *SparsyFed* | Naive PP | *SparsyFed* | Naive PP | *SparsyFed* |
| 0.000 | $84.69 \pm 1.57$ | - | $84.31 \pm 1.01$ | - | $74.86 \pm 2.28$ | - |
| 0.900 | $71.41 \pm 7.01$ | $\mathbf{84.31 \pm 0.86}$ | $66.70 \pm 3.76$ | $\mathbf{82.13 \pm 1.53}$ | $45.82 \pm 6.32$ | $\mathbf{75.00 \pm 2.78}$ |
| 0.950 | $35.02 \pm 6.37$ | $\mathbf{84.25 \pm 1.38}$ | $33.84 \pm 13.71$ | $\mathbf{82.60 \pm 1.58}$ | $24.28 \pm 8.79$ | $\mathbf{75.95 \pm 3.39}$ |
| 0.990 | $12.40 \pm 4.17$ | $\mathbf{77.16 \pm 0.85}$ | $12.30 \pm 3.68$ | $\mathbf{77.71 \pm 1.69}$ | $9.28 \pm 2.52$ | $\mathbf{63.69 \pm 3.90}$ |
| 0.995 | $10.33 \pm 0.57$ | $\mathbf{72.71 \pm 0.65}$ | $10.03 \pm 0.05$ | $\mathbf{70.01 \pm 0.43}$ | $9.74 \pm 3.24$ | $\mathbf{56.79 \pm 3.97}$ |
| 0.999 | $9.86 \pm 0.25$ | $\mathbf{55.24 \pm 2.09}$ | $10.01 \pm 0.02$ | $\mathbf{51.39 \pm 3.19}$ | $11.67 \pm 6.66$ | $\mathbf{43.68 \pm 7.61}$ |

Table 3: Accuracy comparison for Naive Powerpropagation and *SparsyFed* on CIFAR-10 with different LDA settings ($\alpha = 10^3$, $\alpha = 1.0$, and $\alpha = 0.1$).

## E.2    POWERPROPAGATION EXPONENT IN *SparsyFed*

Applying Powerpropagation in *SparsyFed* introduces a new hyperparameter that must be tuned alongside others. To address this, we explored different approaches. The first follows the methodology of the original paper, where a series of fixed exponents were proposed for re-parameterization. In the latter approach, we propose a novel method for determining the exponent dynamically, making it dependent on the network's weights rather than being predefined. This follows a strategy similar to spectral normalization.

### E.2.1    SENSITIVITY ANALYSIS OF FIXED POWERPROPAGATION EXPONENT

Following the approach of the original paper, we evaluated different values of $\beta$ to assess their impact on model performance. Preliminary tests suggest that the sensitivity of this hyperparameter is not as critical as initially expected. We tested several values proposed in the original paper to evaluate their effectiveness in sparse training within a federated learning setting. As shown in Fig. 5, any $\beta$ value between 1 and 2 significantly improves performance in dynamic sparse training compared to the baseline without re-parameterization.

### E.2.2    HYPERPARAMETER-FREE POWERPROPAGATION EXPONENT

To eliminate the need for an additional hyperparameter, we propose an alternative version of *SparsyFed* where $\beta$ is computed at runtime. Instead of a fixed $\beta$ for Powerpropagation, a tailored exponent is derived based on the layer-wise weight magnitude distribution, leveraging concepts from the spectral norm.

We denote the spectral norm of a weight matrix $\mathbf{W}$ as spectral_norm($\mathbf{W}$), which computes the maximum magnitude of the elements in $\mathbf{W}$ and normalizes it by dividing $\mathbf{W}$ by its maximum value $\mathbf{W}_{\max}$. Thus, spectral_norm($\mathbf{W}$) results in a matrix of positive values ranging between 0 and 1. Using spectral_norm($\mathbf{W}$), we construct a tailored exponent matrix for the network's weights by implementing custom convolutional and linear layers, where the matrix is computed at the beginning of the forward pass. The exponentiation process raises each weight $w$ in a layer to the power of

$1 + \text{spectral\_norm}(w)$, ensuring that each weight is scaled by a factor between 1 and 2, proportional to its relative magnitude compared to the largest weight in the network.

To reduce computational overhead, the exponent matrix is calculated only during the first forward pass and stored for subsequent iterations. Additionally, to minimize memory usage, we avoid storing the full exponent matrix and instead compute and store only the average value per layer, reducing memory requirements to a single scalar per layer. This averaging approach results in higher values for denser layers with more nonzero weights, while highly sparse layers tend to have significantly smaller values. The sparsest layers, often found toward the end of the network, tend to be larger and more sparsely populated, amplifying this effect.

The overall computational cost remains marginal and comparable to the standard Powerpropagation implementation, as the computation is only performed during the first forward pass. Performance analysis in Fig. 5 shows that this method outperforms naive `Top-K` pruning without reparameterization and matches or surpasses most fixed alpha values. However, it still falls short of the best-performing fixed $\beta$, suggesting that further refinements would be necessary to bridge this performance gap. While more extensive experiments are needed, this represents a useful insight into potential enhancements for this approach.

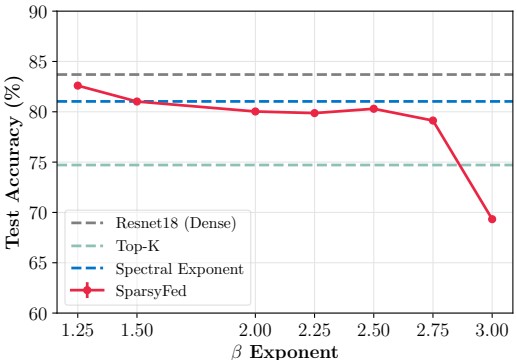

Figure 5: Test Accuracy with different $\beta$ values, with $95\%$ sparsity on CIFAR-10 LDA $\alpha = 1.0$. The accuracy of the dense model (gray), the hyperparameter-free Spectral Exponent version, and the `Top-K` method are also reported for reference.

### E.3 WEIGHT MOVEMENT METRICS IN FEDERATED LEARNING

To better understand the aftermath of *SparsyFed* on the weight of the model during training, a comprehensive evaluation of the weight evolution throughout training has been conducted by comparing the initial global model, the local updates sent to the server at the end of each round, and the subsequently obtained aggregated global model. This allows for gated metrics that capture the overall training progression and the round-specific dynamics. We focus on two primary metrics: the L2 norm between weight matrices and cosine similarity. The former (L2) quantifies the magnitude of weight changes. It is measured as follows: (i) the cumulative L2 norm between the initial model and the aggregated model at each round for long-term evolution, and (ii) the L2 norm between consecutive global models to assess round-level variation. In contrast, cosine similarity captures functional consistency in the updates. It is measured as (i) the similarity between client updates to assess alignment across non-IID data distributions and (ii) between consecutive global models to evaluate directional stability in weight updates.

Experiments were conducted using ResNet18 on CIFAR-10, CIFAR-100 and Speech Commands, with data partitioned using LDA ($\alpha = 0.1$) to simulate non-IID distributions. The model was trained for 200 rounds using *SparsyFed* and `Top-K` under sparsity levels of $95\%$. The results reveal distinct training behaviors across methods.

As shown in Fig. 6, the global L2 distance—measuring the deviation from the initial model—smoothly increases in *SparsyFed*, whereas in other implementations, it rises sharply before stabilizing after several rounds, indicating excessive drift in the early stages. In contrast, *SparsyFed*

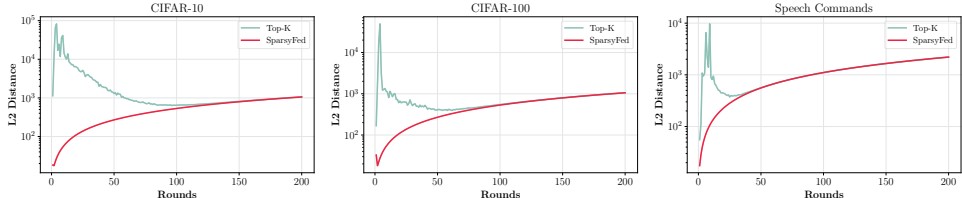

Figure 6: Global L2 Norm – This metric tracks how far the global model moves from its initial state throughout training. A smoother increase indicates more stable updates, while sharp rises suggest rapid drift.

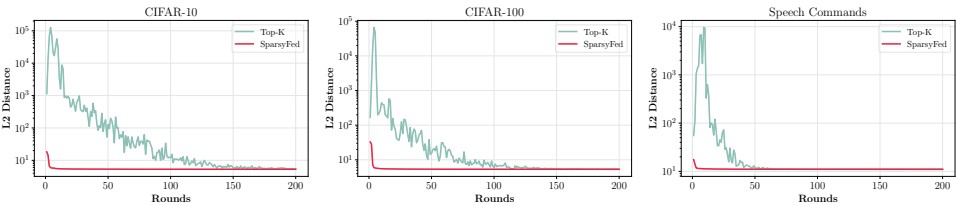

Figure 7: Round L2 Norm – By measuring the L2 norm between consecutive global models, this metric highlights the magnitude of updates applied at each training round, revealing how steadily or abruptly the model evolves.

exhibits a more gradual trajectory, suggesting stable and incremental updates. A similar trend is observed when examining round-by-round evolution, highlighting the steps taken by the model after each aggregation. Fig. 7 illustrates the high variance in updates for `Top-K`, whereas *SparsyFed* maintains stable and smaller updates.

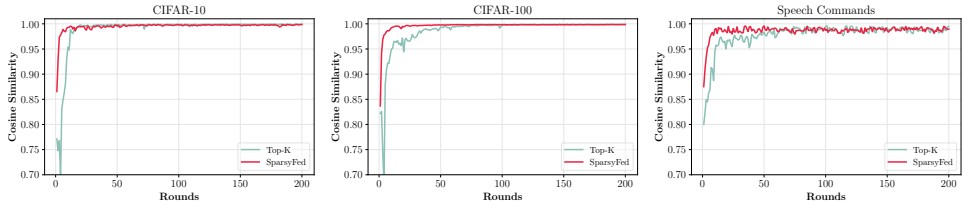

Figure 8: Round Cosine Similarity – This captures the consistency of global model updates across rounds. A high and stable similarity suggests smooth training dynamics, while fluctuations may indicate instability in model aggregation.

Cosine similarity metrics provide complementary insights. The round-wise cosine similarity (Fig. 8) represents the cosine similarity between the global model before local training and after aggregation. This measurement shows how much the model is modified by a training round. The results highlight that *SparsyFed* quickly attains high alignment and maintains stability throughout training, while `Top-K` exhibits more significant fluctuations in the initial rounds, suggesting inconsistent updates across rounds.

Figure 9 shows client round-wise cosine similarity, which measures the similarity among updates sent to the server by clients at the end of each training round. The results indicate that *SparsyFed* tends to have a faster and smoother trajectory. In both cases—client and round— all methods eventually achieve high similarity, though at different stages of training. Notably, *SparsyFed* demonstrates the fastest convergence and smoother dynamics. A slightly lower final similarity value in the CIFAR-10 experiment is not concerning, provided it remains consistent and stable throughout training, as this may indicate better handling of data heterogeneity, allowing clients to adapt more effectively to their respective data distributions.

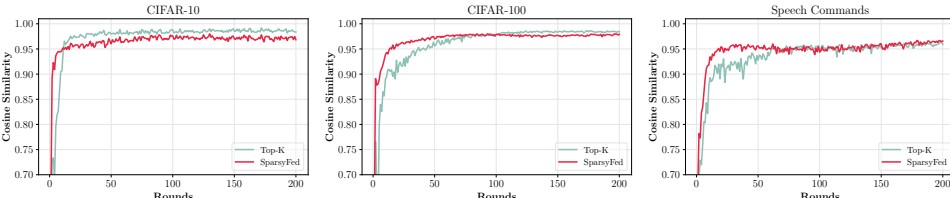

Figure 9: Client Cosine Similarity – This measure assesses how similar client updates are. Higher similarity suggests greater alignment across different clients, while lower values indicate more diverse local updates due to data heterogeneity.

### E.4 GLOBAL MODELS SPARSITY LEVELS

The global model's sparsity fluctuates during training due to the mismatch in client updates. As a result, the global model's sparsity is not always fixed and can fluctuate significantly throughout the training process. The following figures show that the sparsity target directly influences the sparsity level. Higher sparsity targets tend to lead to more significant weight regrowth during training, as seen in Fig. 12, which results in a more considerable mismatch on the server, leading to a denser model. This effect is particularly noticeable at the beginning of training, when the model is more susceptible to significant changes in shape, as illustrated in Fig. 3. Following this, we present a plot of the sparsity measure on the global model for different sparsity targets during training.

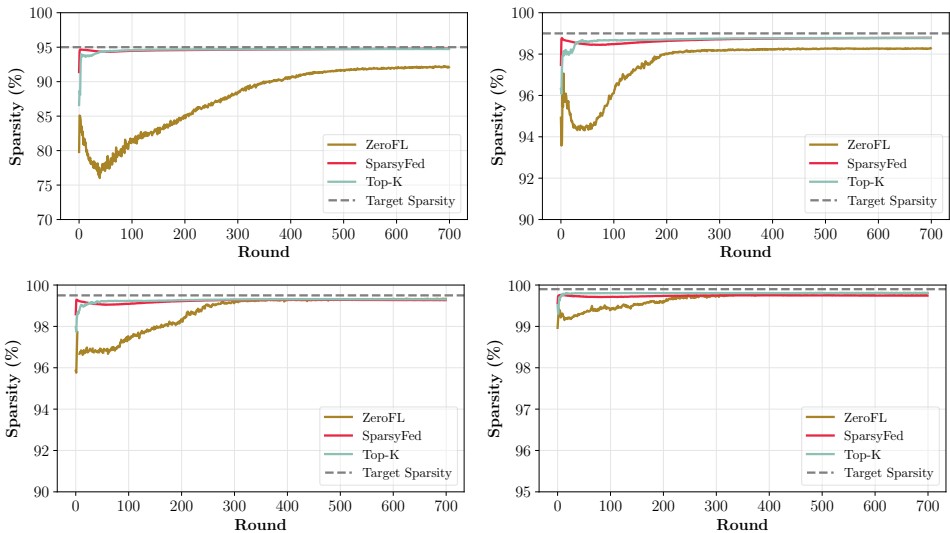

Figure 10: **(a)** From the top left to the bottom right, we present the plots of the sparsity levels for ZeroFL, *SparsyFed*, and Top-K at sparsity levels of 95%, 99%, 99.5%, and 99.9%. ZeroFL struggles to reach the target sparsity in all cases, partially due to its aggregation method, which only aggregates non-zero weights. This leads to large magnitude weights, even if they are present in only a fraction of the clients. *SparsyFed* and Top-K tend to reach the target more quickly, with *SparsyFed* showing a small fluctuation in the initial training phase due to the movement of the mask, as shown in Fig. 3.

### E.5 DISTRIBUTION OF THE SPARSITY THROUGH THE LAYERS.

Sparsity distribution is crucial as the sparsity achieved in the weights of each layer is used to determine the sparsity level applied to the activations during the backward pass. Each layer is pruned with a distinct sparsity level based on the information it contains, leveraging layer sensitivity to implement an effective pruning strategy for activations. This ensures that the sensitivity observed in the weights is reflected in the activation pruning, allowing for dynamic sparsity for both weights

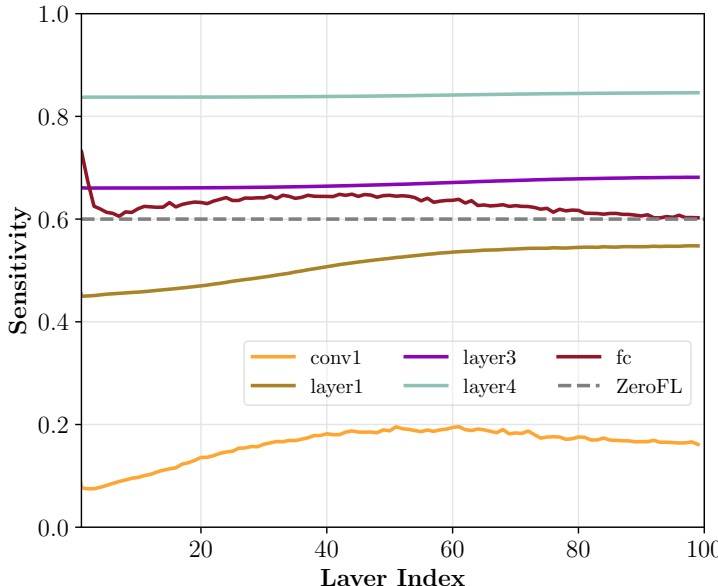

Figure 11: Different sparsity ration of some relevant layer, ZeroFL (red) vs *SparsyFed* with Powerpropagation and sparsity ratio of $60\%$. The first layers are not shown as they are kept dense in ZeroFL. Empirical observation regarding the nature of the pruning procedure (constant across layers for ZeroFL, variable unstructured across layers for *SparsyFed*). We could use this in the main paper only if we make the case that our method performs better because of this. We consider this appendix material for now.

and activations based on the natural sensitivity of each layer. Empirical evidence, shown in Fig. 11, supports this approach, showing that in ZeroFL implementations, the sparsity of the layers remains uniform across all layers of the model, while in others, the sparsity levels vary significantly between layers. The first layers tend to be nearly fully dense, while the deeper layers exceed the global sparsity target, indicating that the first layers retain more information than the deeper layers.

A key factor in this behavior is the limited weight regrowth observed in *SparsyFed*, where only a small number of weights transition from zero to non-zero values after each training round. As depicted in Fig. 12, *SparsyFed* exhibits minimal regrowth, stabilizing quickly over a few rounds. This is directly attributable to using Powerpropagation during training, drastically reducing the impact of smaller weights. Although this behavior was not highlighted in the original paper, it represents empirical evidence supporting the effectiveness of an inherited sparse model training procedure.

As illustrated in Fig. 3, our implementation shows that the global model's mask shifts slightly during the initial training rounds. This suggests that overly rigid approaches, such as those proposed in FLASH, may negatively impact performance by failing to accommodate necessary flexibility. On the other hand, *SparsyFed*'s approach maintains flexibility while still converging toward a consistent global mask.

### E.6 HETEROGENEOUS SPARSITY EXPERIMENTS

In this experimental setup, we aim to evaluate heterogeneous sparsity by using two distinct sets of model sparsity:

1. $[\mathbf{0.9}, \mathbf{0.85}, \mathbf{0.8}]$: These sparsity levels are based on the settings proposed by FLASH in their heterogeneous setup. These values represent a moderate range of sparsity, which does not significantly impact the model's performance in this task.
2. $[\mathbf{0.99}, \mathbf{0.95}, \mathbf{0.9}]$: These denser models previously tested in other experiments offer a more challenging setup. *SparsyFed* outperforms FLASH in these settings due to its ability to adapt efficiently to varying sparsity levels across clients.

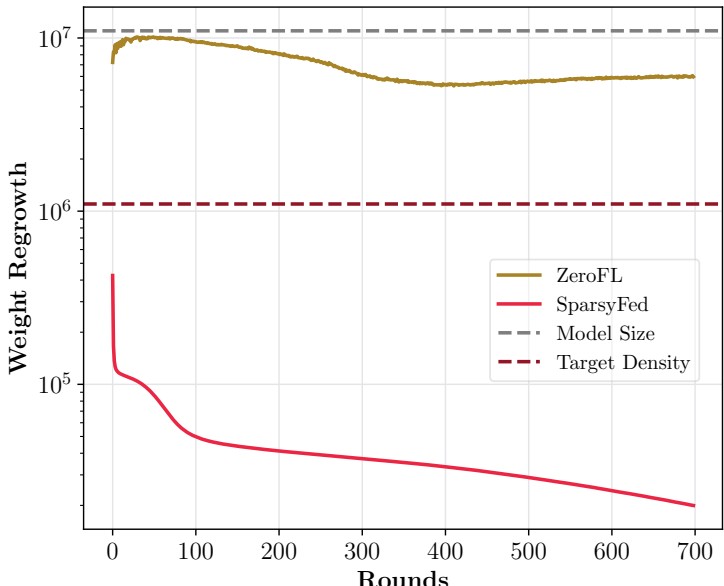

Figure 12: Number of weights regrown (weights switching from zero to non-zero) with one local epoch, compared between *SparsyFed* and ZeroFL. ZeroFL performs the worst of all tested implementations, with $50$-$80\%$ of weights regrowing after local training. In contrast, *SparsyFed* shows near-zero regrowth ($200\times$ lower than ZeroFL). This ensures that the movement of the mask focuses on important weights rather than being distributed across many. **Note**: FLASH does not allow weight regrowth due to its fixed-mask training approach, so it is excluded from this graph.

The clients are partitioned into three groups containing $[40, 30, 30]$ clients. Each group trains on a different level of sparsity in the model. They receive a model matching their capability and send back an update of the same dimensions. For clarity, a client in group 0 in the first setting will receive a model with $0.9$, train on it, and then send back a sparse update with $0.9$. As shown in Table 4, we achieve high performance in all the settings, with a clear margin in the more extreme setting

| Setting | LDA | Lower Sparsity | | Higher Sparsity | |
| | | FLASH | *SparsyFed* | FLASH | *SparsyFed* |
|---|---|---|---|---|---|
| Moderate | $\alpha = 10^3$ | $83.15 \pm 0.96$ | $\mathbf{83.28 \pm 0.44}$ | $83.77 \pm 0.84$ | $83.27 \pm 0.44$ |
| | $\alpha = 0.1$ | $70.9 \pm 0.92$ | $\mathbf{74.97 \pm 2.39}$ | $74.65 \pm 0.96$ | $\mathbf{74.98 \pm 2.39}$ |
| Extreme | $\alpha = 10^3$ | $74.72 \pm 0.49$ | $\mathbf{81.04 \pm 0.26}$ | $74.72 \pm 0.67$ | $\mathbf{81.20 \pm 0.39}$ |
| | $\alpha = 0.1$ | $57.63 \pm 4.83$ | $\mathbf{69.74 \pm 0.28}$ | $59.82 \pm 2.99$ | $\mathbf{69.96 \pm 0.19}$ |

Table 4: Performance comparison between FLASH and *SparsyFed* across different heterogeneous sparsity settings and LDA values. The model has been evaluated on all the sparsity-level trained. For simplicity, we show the test accuracy for the denser model (which reaches a density equal to the target density trained, $0.8$ and $0.95$ for moderate and extreme settings) and the less dense models ($0.9$ and $0.99$ for moderate and extreme). While in the moderate setting, the performances are similar and in line with the dense model's performance (see Table 1), in the extreme setting, *SparsyFed* demonstrated more versatility, achieving high performance even with the sparsest model. It is important to note that in such a setting, the model's performance would align with the sparsest model trained, especially considering it is trained from a larger (though not majority) group of clients compared to the others.

## E.7 WIDE FEDERATED CIFAR10

In this experiment, we aimed to address the challenge of training a model in a setting where the number of samples per client is extremely low. To do this, following an approach similar to the one proposed in Charles et al. (2021a), we partitioned the CIFAR-10 dataset into 1000 clients following an LDA distribution with $\alpha = 0.1$. As shown in Table 5, *SparsyFed* significantly outperformed the alternative in this setting, where clients retain minimal information. Notably, both sparse methods outperformed the dense one in this setting, likely due to the dilution of information in the dense model when such a small amount of data is used at each round.

| Method | Accuracy (%) |
|---|---|
| Dense | $47.11 \pm 2.77$ |
| FLASH | $51.96 \pm 2.84$ |
| *SparsyFed* | $\mathbf{54.37 \pm 1.28}$ |

Table 5: Performance comparison of the dense model, FLASH, and *SparsyFed*. Both FLASH and *SparsyFed* have been trained with a sparsity of 0.95. The experiments were conducted on CIFAR-10, partitioned across 1000 clients following LDA with $\alpha = 0.1$, and a participation rate of 0.1%.

## E.8 VISION TRANSFORMER

To extend our experimental evaluation, we designed an experiment using a Vision Transformer (ViT) (Dosovitskiy et al., 2021) on the CUB-200-2011 dataset (Welinder et al., 2010), following the setup proposed in (Hu et al., 2023). Specifically, we use a pre-trained ViT-Base model on ImageNet-21k (Deng et al., 2009) and fine-tuned on CIFAR-100.[1] The data is partitioned using LDA with $\alpha = 1000.0$ among 100 clients, each receiving approximately 60 samples. The participation rate for training and evaluation is set to 10%. The experiments were conducted using the AdamW optimizer with various settings. While hyperparameter tuning can be challenging and resource-intensive in this context, initial results indicate a noticeable advantage of using *SparsyFed* over a naive `Top-K` approach in federated learning. The following results compare four different combinations of learning rate and the number of local epochs. In the first case, only one local epoch is performed per round, while in the other cases, three local epochs are used, with the first two serving as a warm-up phase. Additionally, two different learning rate strategies are evaluated: one with a fixed learning rate of 0.01 and another with a server-side decay schedule that reduces the learning rate from 0.01 to 0.001. The global sparsity is set to 50%, and the *SparsyFed* and `Top-K` implementations are compared. Overall, *SparsyFed* leads to a consistent improvement in performance, with more evident benefits in some instances. In all settings, *SparsyFed* can remarkably maintain a sparsity level close to the target throughout training. In contrast, the `Top-K` implementation tends to drift away from the desired sparsity as training progresses. Our method thus reduces communication overhead during training while effectively achieving the target sparsity level in the final model.

---

[1]The pre-trained model was sourced on the Hugginface Hub from this repository

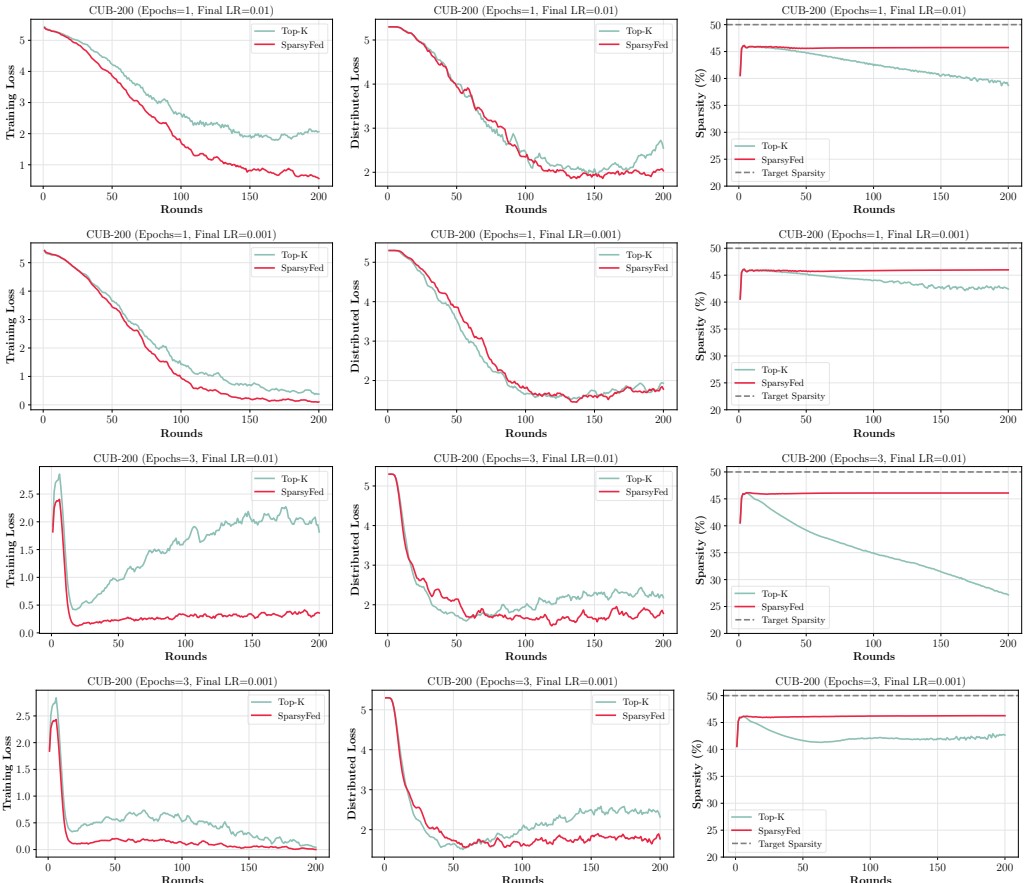

Figure 13: Experiment using a Vision Transformer (ViT) on the CUB-200-2011 dataset with LDA $\alpha = 1000.0$, 100 clients, and 10% partial participation rate for training and evaluation. For each experimental setting (rows), we show the training loss, distributed evaluation loss, and sparsity level on the left, central, and right columns, respectively. The training loss is measured during training. The distributed evaluation loss is collected during the client data evaluation. The sparsity level refers to the global model after aggregation on the server. Different settings are shown, from top to bottom: (first row) one local epoch with a fixed learning rate, (second row) one local epoch with server-side learning rate decay from 0.01 to 0.001, (third row) three local epochs with a fixed learning rate, and (fourth row) three local epochs with server-side learning rate decay from 0.01 to 0.001. The number of local epochs and the final learning rate are reported at the top of each plot.

## E.9 FULL TABLES ACCURACY RESULTS

| Dataset | Sparsity | Resnet18 | ZeroFL | FLASH | *SparsyFed* |
|---|---|---|---|---|---|
| CIFAR10 ($\alpha = 10^3$) | dense | $85.14 \pm 1.18$ | - | - | - |
| | 0.9 | $82.16 \pm 0.80$ | $78.67 \pm 1.52$ | $82.57 \pm 2.05$ | $\mathbf{84.31 \pm 0.86}$ |
| | 0.95 | $77.92 \pm 0.97$ | $76.16 \pm 1.28$ | $82.22 \pm 0.14$ | $\mathbf{84.25 \pm 1.38}$ |
| | 0.99 | $68.11 \pm 3.50$ | $72.40 \pm 1.08$ | $\mathbf{77.48 \pm 2.86}$ | $77.16 \pm 0.85$ |
| | 0.995 | $54.49 \pm 8.70$ | $60.31 \pm 4.11$ | $71.85 \pm 1.63$ | $\mathbf{72.71 \pm 0.65}$ |
| | 0.999 | $17.87 \pm 6.22$ | $21.91 \pm 1.11$ | $46.43 \pm 16.73$ | $\mathbf{55.24 \pm 2.09}$ |
| CIFAR100 ($\alpha = 10^3$) | dense | $53.79 \pm 1.63$ | - | - | - |
| | 0.9 | $46.47 \pm 1.74$ | $44.11 \pm 0.93$ | $53.06 \pm 1.10$ | $\mathbf{54.44 \pm 1.33}$ |
| | 0.95 | $13.45 \pm 21.45$ | $33.67 \pm 3.34$ | $49.25 \pm 1.63$ | $\mathbf{54.33 \pm 1.48}$ |
| | 0.99 | $1.54 \pm 1.22$ | $10.59 \pm 2.48$ | $44.40 \pm 1.77$ | $\mathbf{47.62 \pm 1.67}$ |
| | 0.995 | $0.97 \pm 0.64$ | $4.04 \pm 3.43$ | $36.82 \pm 1.72$ | $\mathbf{42.05 \pm 1.21}$ |
| | 0.999 | $0.97 \pm 0.64$ | $0.87 \pm 0.21$ | $9.61 \pm 3.61$ | $\mathbf{13.85 \pm 1.01}$ |
| Speech Commands ($\alpha = 10^3$) | dense | $92.98 \pm 0.67$ | - | - | - |
| | 0.9 | $85.53 \pm 0.84$ | $89.12 \pm 1.11$ | $89.89 \pm 0.74$ | $\mathbf{91.34 \pm 0.52}$ |
| | 0.95 | $80.19 \pm 1.92$ | $85.63 \pm 1.12$ | $88.16 \pm 1.37$ | $\mathbf{89.79 \pm 0.21}$ |
| | 0.99 | $67.67 \pm 2.32$ | $60.79 \pm 2.44$ | $76.41 \pm 1.31$ | $\mathbf{77.00 \pm 0.73}$ |
| | 0.995 | $36.69 \pm 1.45$ | $41.16 \pm 3.10$ | $67.54 \pm 1.15$ | $\mathbf{70.03 \pm 1.14}$ |
| | 0.999 | $63.63 \pm 3.24$ | $16.77 \pm 6.23$ | $31.83 \pm 2.18$ | $\mathbf{49.27 \pm 0.50}$ |

Table 6: Aggregated results for CIFAR10, CIFAR100, and Google Speech Command datasets.

| Dataset | Sparsity | Resnet18 | ZeroFL | FLASH | *SparsyFed* |
|---|---|---|---|---|---|
| CIFAR10 ($\alpha = 1.0$) | dense | $83.70 \pm 1.70$ | - | - | - |
| | 0.9 | $80.56 \pm 1.90$ | $76.16 \pm 1.30$ | $81.15 \pm 1.03$ | $\mathbf{82.13 \pm 1.53}$ |
| | 0.95 | $74.71 \pm 3.29$ | $75.53 \pm 2.27$ | $79.36 \pm 1.03$ | $\mathbf{82.60 \pm 1.58}$ |
| | 0.99 | $66.27 \pm 5.08$ | $70.71 \pm 0.15$ | $73.45 \pm 1.37$ | $\mathbf{77.71 \pm 1.69}$ |
| | 0.995 | $63.82 \pm 2.41$ | $56.02 \pm 3.95$ | $69.15 \pm 1.60$ | $\mathbf{70.01 \pm 0.43}$ |
| | 0.999 | $31.79 \pm 19.10$ | $17.66 \pm 8.34$ | $36.07 \pm 7.49$ | $\mathbf{51.39 \pm 3.19}$ |
| CIFAR100 ($\alpha = 1.0$) | dense | $52.29 \pm 1.14$ | - | - | - |
| | 0.9 | $46.57 \pm 1.71$ | $40.70 \pm 4.72$ | $51.99 \pm 0.21$ | $\mathbf{53.08 \pm 0.90}$ |
| | 0.95 | $28.07 \pm 23.27$ | $38.82 \pm 1.75$ | $47.19 \pm 1.88$ | $\mathbf{52.81 \pm 1.72}$ |
| | 0.99 | $19.65 \pm 16.30$ | $18.97 \pm 2.08$ | $42.76 \pm 4.08$ | $\mathbf{46.64 \pm 1.59}$ |
| | 0.995 | $9.51 \pm 14.81$ | $6.01 \pm 4.74$ | $36.43 \pm 4.97$ | $\mathbf{42.21 \pm 1.03}$ |
| | 0.999 | $3.81 \pm 2.18$ | $1.96 \pm 0.66$ | $5.80 \pm 2.86$ | $\mathbf{15.96 \pm 0.64}$ |
| Speech Commands ($\alpha = 1.0$) | dense | $91.49 \pm 0.94$ | - | - | - |
| | 0.9 | $84.28 \pm 0.88$ | $87.79 \pm 1.40$ | $88.68 \pm 1.72$ | $\mathbf{92.32 \pm 1.59}$ |
| | 0.95 | $78.58 \pm 0.44$ | $84.29 \pm 1.50$ | $84.89 \pm 0.49$ | $\mathbf{89.14 \pm 1.15}$ |
| | 0.99 | $65.01 \pm 0.84$ | $57.79 \pm 0.82$ | $69.22 \pm 1.59$ | $\mathbf{75.82 \pm 3.72}$ |
| | 0.995 | $56.73 \pm 1.00$ | $37.16 \pm 2.71$ | $58.23 \pm 1.84$ | $\mathbf{68.02 \pm 3.14}$ |
| | 0.999 | $21.56 \pm 12.79$ | $10.10 \pm 4.01$ | $17.70 \pm 2.58$ | $\mathbf{47.43 \pm 1.66}$ |

Table 7: Aggregated results for CIFAR10, CIFAR100, and Google Speech Command datasets.

| Dataset | Sparsity | Resnet18 | ZeroFL | FLASH | *SparsyFed* |
|---|---|---|---|---|---|
| CIFAR 10 ($\alpha = 0.1$) | dense | $73.81 \pm 4.84$ | - | - | - |
| | 0.9 | $69.79 \pm 3.78$ | $67.40 \pm 4.11$ | $71.87 \pm 2.63$ | $\mathbf{75.00 \pm 2.78}$ |
| | 0.95 | $60.00 \pm 4.66$ | $61.55 \pm 4.18$ | $72.08 \pm 2.09$ | $\mathbf{75.95 \pm 3.39}$ |
| | 0.99 | $43.96 \pm 11.99$ | $51.71 \pm 3.54$ | $56.91 \pm 3.55$ | $\mathbf{63.69 \pm 3.90}$ |
| | 0.995 | $19.02 \pm 10.77$ | $41.33 \pm 3.64$ | $52.15 \pm 3.87$ | $\mathbf{56.79 \pm 3.97}$ |
| | 0.999 | $11.5 \pm 4.494$ | $18.76 \pm 4.28$ | $29.31 \pm 6.75$ | $\mathbf{43.68 \pm 7.61}$ |
| CIFAR100 ($\alpha = 0.1$) | dense | $48.34 \pm 2.71$ | - | - | - |
| | 0.9 | $41.96 \pm 2.16$ | $31.92 \pm 7.65$ | $45.59 \pm 0.75$ | $\mathbf{48.37 \pm 1.73}$ |
| | 0.95 | $11.48 \pm 17.51$ | $34.21 \pm 7.65$ | $44.31 \pm 2.14$ | $\mathbf{48.27 \pm 2.70}$ |
| | 0.99 | $0.14 \pm 0.72$ | $13.07 \pm 2.26$ | $34.75 \pm 3.38$ | $\mathbf{41.03 \pm 2.14}$ |
| | 0.995 | $0.14 \pm 0.72$ | $7.04 \pm 5.25$ | $26.44 \pm 17.35$ | $\mathbf{35.72 \pm 2.01}$ |
| | 0.999 | $0.14 \pm 0.72$ | $1.66 \pm 0.97$ | $3.56 \pm 2.07$ | $\mathbf{13.84 \pm 3.69}$ |
| Speech Commands ($\alpha = 0.1$) | dense | $80.15 \pm 2.69$ | - | - | - |
| | 0.9 | $65.44 \pm 0.97$ | $70.35 \pm 2.65$ | $77.15 \pm 0.77$ | $\mathbf{79.67 \pm 2.78}$ |
| | 0.95 | $57.39 \pm 1.04$ | $65.90 \pm 1.88$ | $71.28 \pm 1.75$ | $\mathbf{75.46 \pm 2.24}$ |
| | 0.99 | $50.42 \pm 6.26$ | $41.42 \pm 1.60$ | $53.55 \pm 2.00$ | $\mathbf{56.69 \pm 4.56}$ |
| | 0.995 | $34.20 \pm 1.43$ | $22.61 \pm 3.45$ | $43.16 \pm 3.47$ | $\mathbf{48.30 \pm 5.39}$ |
| | 0.999 | $19.25 \pm 6.01$ | $8.85 \pm 3.76$ | $17.14 \pm 2.97$ | $\mathbf{29.24 \pm 2.34}$ |

Table 8: Aggregated results for CIFAR10, CIFAR100, and Google Speech Command datasets.

## F    ADDITIONAL EXPERIMENTAL CONFIGURATION DETAILS

**Learning rate scheduler.** The learning rate follows a scheduled pattern defined by the function:

$$\eta_t = \eta_{\text{start}} \exp\left( \frac{t}{T} \ln\left( \frac{\eta_{\text{end}}}{\eta_{\text{start}}} \right) \right) \tag{1}$$

**Reproducibility.** Seeds were used for client sampling, while others were fixed for reproducibility purposes. All simulations were conducted using three different client sampling seeds: 5378, 9421, and 2035.

**Experimental Setting.** Each round consisted of one local epoch with a local batch size of 16 samples. The initial learning rate was set to 0.5, gradually decreasing to a final value of 0.01 following Eq. (1). For *SparsyFed*, the exponent for re-parameterization was set to $\beta = 1.25$ for the CIFAR-10/100 experiments and $\beta = 1.15$ for the Speech Commands experiment. The CIFAR experiments were run for 700 rounds, while the Speech Commands experiment was run for 500 rounds.

## G    COMPARATIVE ANALYSIS ON ALGORITHMS AND BASELINES

Here is a brief description of the implementation used during the experiments for Top-K, ZeroFL Qiu et al. (2022), FLASH Babakniya et al. (2023), and *SparsyFed*:

### G.1    TOP-K

1. **Pruning**: The model is pruned per round at clients after executing the (local) training on their own data, i.e., just before sending the updated model to the server. The pruning method used is global unstructured Top-K, which prunes all the model parameters except for the $k$ largest values.
2. **Aggregation strategy**: FedAvg. Alternatively, other aggregation strategies acting on the pseudo-gradients can also be applied straightforwardly.

### G.2 ZEROFL

1. **Pruning**: Pruning is performed during the local training at clients and before sending the model update back to the central server. During training, the SWAT unstructured (per-layer, in contrast to standard global unstructured `Top-K`) approach is used to prune the weights before the forward pass and the activations during the forward pass. This alone doesn't ensure obtaining a model with the targeted degree of sparsity because SWAT often results in weight regrowth per optimizer step. To achieve the target sparsity, the model is pruned again before sending it back to the server, using the same approach as `Top-K`, i.e., global unstructured. In the original work, three levels of masks ($[0.0, 0.1, 0.2]$) have been proposed to increase the density of the model before the uplink communication. For our experiments, we used the smaller one ($0.0$) since it is more in line with

2. **Aggregation strategy**: A slight variation of FedAvg is used, where the averaging is executed among the non-zero weights to avoid excessive dilution of information in the presence of highly sparse models. Adapting the aggregation function to act only on non-zero weights supports alternative aggregation strategies acting on the pseudo-gradients.

### G.3 FLASH - SPDST (SENSITIVITY-DRIVEN PRE-DEFINED SPARSE TRAINING)

(THE ONE USED IN THE EXPERIMENTS)

1. **Pruning**: Pruning is performed at the end of the first round of training (similar to `Top-K`), which, in the original paper, is referred to as a warm-up phase. Thus, the first round of training is executed using the dense model, producing the initial mask. The binary mask obtained during the warm-up phase is fixed for the subsequent federated rounds, and only the non-zero weights are trained. This means there is no need for further pruning of the weights in subsequent rounds since no regrowth is allowed (in this version of Flash).

2. **Aggregation strategy**: The aggregation is performed only among the non-zero weights, similar to ZeroFL. During the first aggregation, at the end of the initial training round, the model is further pruned on a per-layer basis to counter the regained density caused by the mismatch in local masks. The pruning uses the average sparsity level all clients achieve for each layer. For example, if the average sparsity of layer $l$ is $d_l$, adjusted by a factor $r$, then layer $l$ of the pruned global model will have sparsity $d_l$. The factor $r$ helps to maintain the target global sparsity by ensuring that the sum of individual layer sparsity levels meets the overall goal. The resulting binary mask becomes the final one, preserved for all subsequent training rounds. This process, defined in the original paper as sensitivity analysis, is applied only in the first round, as fixed mask training prevents further mask modification in later rounds.

### G.4 FLASH - JMWST (JOINT MASK WEIGHT SPARSE TRAINING)

1. **Pruning**: A normal training procedure is applied (NO FIXED MASKS). Pruning is performed at the end of each local training session.

2. **Aggregation strategy**: The server aggregates and then prunes the model, applying the same sensitivity analysis introduced in SPDST to address the regained density. This is done every $r$ round, as the training method allows for the regrowth of the clients. The original paper proposed two values for $r$: $r = 1$ and $r = 5$.

### G.5 *SparsyFed*

1. **Pruning**: Powerpropagation is applied to re-parameterize the weights during the forward pass executed at clients. Pruning during training is applied only to the activations during the backward pass. At the end of the local training, the model is pruned and returned to the server. The first round of training executes using a full-size model.

2. **Aggregation strategy**: FedAvg. Alternatively, other aggregation strategies acting on the pseudo-gradients can also be applied straightforwardly.

## H  FLOPS REDUCTIONS FOR UNSTRUCTURED SPARSE TRAINING.

Unstructured sparsity can theoretically reduce FLOPs linearly in the percentage of zero weights (Dettmers & Zettlemoyer, 2019; Singh et al., 2024) if an optimal sparse algorithm exists for a given operation. However, such gains are fully realized only when the hardware can efficiently skip zero-valued operations. In contrast, structured sparsity methods—such as block-sparsity pruning (Gray et al., 2017)—yield more predictable speed-ups on standard GPUs due to optimized kernels despite typically achieving lower overall sparsity than unstructured sparsity. Emerging accelerators and libraries increasingly support unstructured sparsity (Jeong et al., 2024), bridging the gap between theoretical FLOP reductions and actual runtime improvements.

