# OpenReview forum: "SparsyFed: Sparse Adaptive Federated Learning"
_ICLR.cc/2025/Conference — ICLR 2025 Poster_

### Official Review · Reviewer_75sH · 2024-10-30

**Soundness:** 2
**Presentation:** 2
**Contribution:** 2
**Rating:** 3
**Confidence:** 3

**Summary:**

This paper proposes a sparse FL method to tackle the following problems in FL: 1) data heterogeneity and its impacts on sparsity across clients; 2) adapt to never-seen data distributions; 3) hyperparameter turning in FL. The authors present *SparsyFed* and claim that it can produce very sparse models, while offering *plasticity* to data distributions.

**Strengths:**

The authors conduct comprehensive empirical evaluations of the proposed methods. In particular, the proposed method is tested on multiple datasets with different data distributions. Besides model accuracy, the authors also analyze reductions in communication costs when using the sparse training method.

In addition, the authors also conduct necessary ablation studies to investigate 1) sparsity pattern dynamics during training; 2) different weight re-parametrization methods; 3) activation pruning.

**Weaknesses:**

**1 Claims are not fully supported**: The authors claim that *SparsyFed* addresses all three problems mentioned in the abstract: data heterogeneity and its impact on clients' sparsity, a lack of sufficient plasticity to never-seen data distributions, and hyperparameter tuning. However, it is unclear that what techniques in *SparsyFed* address these problems. In particular, it is not convincing that consensus in sparsity is achieved by simply adopting a weight re-parameterization scheme. I believe more analyses are needed to support the claim.

In addition, I am also confused about how *SparsyFed* can `adapt to never-seen data distributions`, as there is no technique used in the method that can specifically address this problem.

**2 Presentations in the introduction section**: I did not see any part in the introduction explaining technical details in *SparsyFed*. The authors mainly focus on describing the benefits of *SparsyFed*. However, it is confusing what techniques in *SparsyFed* bring these benefits.

**3 Contributions are not fully explained in Sec 3**:

First, I am confused about lines 6-10 in Algo 2. Some functions like *GetLayer* and *SetLayer* are not explained. As these lines are the main contribution to the paper, it is hard to understand why *SparsyFed* achieves good performance without knowing its technical details.

Furthermore, the authors did not explain why sparse activations help attain a sparse model. As it is not a theoretical work, more intuitive explanations and evaluations are needed to justify such operations.

**Questions:**

See the comments above.

---

> ### Author Response · Authors · 2024-11-25
>
> Dear Reviewer 75sH,
>
> Thank you for your detailed and constructive comments. Below, we address each of your concerns in detail:
>
> 1. **Supporting claims**:
>     - We acknowledge that some results and explanations may have seemed unclear. To address this, we have revised the Introduction and Evaluation sections, particularly those related to sparsity consensus (Section 5.3), and expanded the ablation studies of our algorithm in Sections 5.4 and 5.5. Specifically, the weight reparameterization (5.4) plays a critical role, as it inherently induces sparsity in the model while maintaining sparsity resilience during training. This approach ensures that clients train on the same subspace of the global model, improving consensus on the sparse masks. For example, weight reparametrization allows SparsyFed to reach convergence 7-8x times faster, as shown in Fig. 1.
> 2. **Introduction**:
>     - To clarify *SparsyFed'*s technical contributions, we have restructured the introduction to emphasize the specific mechanisms we used. These include weight reparameterization and sparse activation pruning. We also expand upon how our layer-wise activation pruning at the end of each round significantly differs from previous works.
>         - Unlike previous methods, which applied fixed global sparsity across layers, our layer-wise approach prunes based on each layer's proportion of total parameters. This removes more capacity from dense layers while preserving parameter-efficient ones.
>         - Pruning models at the end of each round allows full flexibility for local optimization. By starting from a dense model in the first round, SparsyFed also provides robust initialization for crucial layers, such as embeddings or class projectors. By contrast, earlier works which pruned from the first step had to exclude layers to preserve performance.
>     - We emphasize that SparsyFed only introduces one new hyperparameter, the $\beta$ term in our weight reparameterization, making it suitable for tuning under the time-intensive Federated Learning paradigm.
> 3. **Algorithm clarity**:
>     - To enhance the clarity of Algorithm 2, we have extended the explanations for its key steps, in ‘Pruning Activations During Local Training’ and 'Pruning Activations During Local Training’, and introduced a pipeline diagram to visually represent the workflow (New Fig.1).
>     - Specifically:
>         - At **line 7**, the `GetLayer` function extracts the weights and activations for the current layer.
>         - At **line 8**, the extracted weights are used to determine the pruning levels for pruning the model.
>         - At **line 9**, the activations are pruned using the defined pruning thresholds.
>         - At **line 10**, the `SetLayer` function updates the sparse weights and activations in the model.
>         - At **line 11**, the sparse activations and weights are used to compute gradients, ensuring that sparsity is preserved throughout training.
>     - We want to emphasize that we do not use sparse activations to obtain a sparse model, instead we use them to improve computational efficiency. This has been clarified in Section 5.5. The efficiency benefits of using sparse activations have been cataloged in previous works in a centralized environment[1] and in a federated context[2]. In [1] they registered 5.6x speed-up for the forward pass and a 6.3x sped-up for the backward pass on appropriate hardware.
>
> We hope these revisions address your concerns and clarify the key contributions of our work. Thank you for your valuable feedback, which has helped us improve the quality and presentation of our manuscript.
>
> Sincerely,
>
> The Authors
>
> [1] Raihan, Md Aamir, and Tor M Aamodt. "Sparse weight activation training." In Advances in Neural Information Processing Systems, December 2020.
>
> [2] Qiu, Xinchi, Javier Fernandez-Marques, Pedro PB Gusmao, Yan Gao, Titouan Parcollet, and Nicholas Donald Lane. "Zerofl: Efficient on-device training for federated learning with local sparsity." In International Conference on Learning Representations, 2021.

---

> > ### Comment · Reviewer_75sH · 2024-11-27
> > **Response**
> >
> > Thanks for the rebuttal.
> >
> > - Regarding the weight re-parameterization and its effects on sparsity consensus:
> >
> >   I am still not convinced how it works. I understand that the authors provided studies on Sec 5.3. However, These studies did not show how weight re-parameterization leads to sparsity consensus.
> >
> >   Besides, the authors claim one of the goals of this paper is to mitigate tuning works on hyperparameters. However, *SparsyFed* introduces additional parameters such as weight re-parameterization exponent, $beta$. And it looks tricky to determine its value (Sec E.2). Furthermore, I believe it is also necessary to study how $beta$ affects the sparse consensus in different models and datasets.
> >
> > - Regarding the claim of plasticity to adapt to never-seen data distributions:
> >
> >   I am still confused about this claim. Please clarify which part of the proposed method leads to the claim.
> >
> > - Regarding the sparse activation and its effects on computation efficiency:
> >
> >   I think it is necessary to conduct some real experiments and see how the wall-clock training time is improved. Otherwise, evaluating how much benefit it brings and why it is needed is unclear.

---

> > > ### Author Response · Authors · 2024-11-28
> > >
> > > Dear Reviewer 75sH,
> > >
> > > Thank you for responding to our comments. We understand your concerns and sincerely appreciate your feedback. We value your input and aim to improve our work and provide clarity in addressing these issues.
> > >
> > > - **Weight Re-parametrization**:
> > >     - Our intuition is grounded in the literature of the original work [1] and supported by results related to global model sparsity (Fig.2-right), binary mask overlap (Fig.3), and weight regrowth (Section E.4 of the appendix, Fig.8).  We would like to ask if there is a specific evaluation you would recommend to further demonstrate this.
> > > - **Model Plasticity**:
> > >     - The claim regarding model plasticity comes from the effectiveness of the weight re-parameterization technique demonstrated in continuous learning contexts in the original work [1], as well as its observed ability to dynamically adjust the binary mask of the model, unlike fixed-mask approaches.
> > >     - If there is a specific evaluation you would like us to perform to better demonstrate this feature, we would be glad to implement it.
> > > - **Computation efficiency**:
> > >     - We understand that simply pointing out previous literature may not be enough to address the claim. However, evaluating speedups on traditional hardware using standard frameworks such as PyTorch remains challenging, as they are primarily designed for dense matrices. We are actively looking for a framework for an effective evaluation of the effective speedup due to the reduced computation.
> > >     - If you have any recommendations for evaluation frameworks or metrics you would like us to consider, we would be happy to integrate them into our work.
> > >
> > > Thank you once again for your response and the constructive feedback. We are committed to addressing these points comprehensively and welcome any further suggestions you may have.
> > >
> > > Sincerely,
> > >
> > > The Authors
> > >
> > > [1] Jonathan Schwarz et al., "Powerpropagation: A sparsity inducing weight reparameterisation," in 35th Conference on Neural Information Processing Systems (NeurIPS), 2021.

---

### Official Review · Reviewer_Ay19 · 2024-11-03

**Soundness:** 3
**Presentation:** 2
**Contribution:** 2
**Rating:** 5
**Confidence:** 5

**Summary:**

This work presents SparsyFed, an adaptive sparse training method tailored for cross-device federated learning. Authors show that SparsyFed can achieve impressive sparsity levels while minimizing the accuracy drop due to the compression. SparsyFed outperforms in accuracy three federated sparse training baselines, TopK, ZeroFL, and FLASH, using adaptive and fixed sparsity for three typical datasets used in cross-device FL.

**Strengths:**

1.The authors summarize three challenges when process sparse training in cross-device federated learning environments.

2.The authors propose the SparsyFed method to accelerate on-device training in cross-device FL.

3.Extensive experiments show the effectiveness of SparsyFed, which can target high sparsity levels (up to 95%) without compromising the model’s accuracy with a novel approach leveraging hyperparameter-less local activation pruning and thoughtful weight reparameterization.

**Weaknesses:**

1.It’s suggested that authors give a comprehensive survey on adaptive sparse training method. Although authors claim “Previous works have only managed to solve one, or perhaps two of these challenges”, can authors give a comprehensive comparison of existing methods?

2.Considering different clients train different submodels, the server also maintains a full model. So can the sparsity of clients be different to apply for heterogeneous hardware?

3.Can authors further explain why clients should achieves consensus on the clients’ sparse model masks when server always maintain a full model.

4.What’s the definition of the model plasticity?

5.In experimental section, authors only compared with two baselines, there’re several works also focus on the same questions, for example [1,2,3], so it’s suggested to add more experimental to show the effectiveness of proposed method.

6.Considering the model architecture, authors only show the effectiveness on convolutional network, what’s the performance on other architecture, for example Transformer?

[1]Stripelis, Dimitris, et al. "Federated progressive sparsification (purge, merge, tune)+." arXiv preprint arXiv:2204.12430 (2022).

[2]Wang, Yangyang, et al. "Theoretical convergence guaranteed resource-adaptive federated learning with mixed heterogeneity." Proceedings of the 29th ACM SIGKDD Conference on Knowledge Discovery and Data Mining. 2023.

[3]Zhou, Hanhan, et al. "Every parameter matters: Ensuring the convergence of federated learning with dynamic heterogeneous models reduction." Advances in Neural Information Processing Systems 36 (2024).

**Questions:**

See weaknesses.

---

> ### Author Response · Authors · 2024-11-25
>
> Dear Reviewer Ay19,
>
> We greatly appreciate your thorough review and insightful suggestions. Below, we address each of your comments in detail:
>
> 1. **Comprehensive survey on adaptive sparse training methods**:
>     - We acknowledge the need for a more detailed comparison of existing methods. To address this, we revisited the evaluation in Section 5 to ensure that all research questions posed in the initial guidelines were thoroughly answered. For example, we have provided comprehensive ablation studies on the effectiveness of weight reparametrization to stabilize the sparse training and the effectiveness of activation pruning for reducing the total computation cost of the training. Additionally, we expanded the discussion on the importance of consensus and sparsity pattern dynamics in Section 5.3, highlighted the significance of the weight reparameterization method in Section 5.4, and emphasized its effective decoupling with activation pruning in Section 5.5.
> 2. **Global model sparsity**:
>     - In the revised manuscript, in Section 3 we clarified that the server maintains a sparse model for all rounds after the initial round. At each training round, all clients receive the same (sparse) global model, train it locally on their data, and send back sparse local updates. The sparsity of the global model after aggregation will depend on the outer optimizer and the sparsity pattern across clients; however, our weight reparametrization in SparsyFed tends to create consensus on the sparse masks of the clients, limiting interference during aggregation and allowing us to maintain a higher degree of sparsity than the baselines (see the right-side of Fig. 2).
>     - This behaviour is illustrated in Fig. 1(b), where we have updated the caption for greater clarity.
> 3. **Model plasticity**:
>     - A clear definition of model plasticity has been added to the Introduction section. Specifically, plasticity refers to the model’s ability to adapt to new data while retaining previously learned knowledge. Studies in supervised and reinforcement learning have shown that reduced plasticity can lead to overfitting to limited data and hinder generalization[2,3]. This definition has been contextualized to the contrast between fixed-mask approaches which limit plasticity, used by previous works, and the dynamic approach of SparsyFed. As an extreme example, in a multi-task or continual learning setting, neural networks can be iteratively pruned to build task-specific sub-networks[1], which, as an algorithm, requires dynamic masking.
> 4. **Expanding baselines**:
>     - We appreciate the suggestion to include additional baselines. After reviewing the works in [4, 5, 6], we discuss their relevance to our work:
>         - **FedSparsify [4]**: Our approach begins at the target sparsity level. In contrast, FedSparsify gradually increases sparsity over training rounds, meaning it spends a significant chunk of training communicating the dense model and thus lacks the crucial communication-reduction benefits of SparsyFed. For example, we obtain a 10x reduction in total up and down-link communication costs while the best performing **FedSparsify** method obtains a 3-4x reduction~(See Table 1.a and Table 1.b of [4]).
>         - **RA-Fed [5]**: This work focuses on training a dense global model in resource-constrained settings rather than a sparse model. Thus, it deviates from the goals of our work, which are (a) to train a sparse model and (b) to reduce communication costs during federated training. As such, we do not consider it a relevant baseline.
>         - **Zhou et al. [6]**: The goal of this work is not to propose a new competitive sparsification method; rather, it proposes a framework for analyzing the rate of convergence of sparse training methods. We thank the reviewer for pointing out the paper to us, and we are looking into how we could apply a similar analysis to our work. However, it does not represent an experimental baseline for SparsyFed.
>
> We hope these revisions address your concerns comprehensively. Thank you again for your constructive feedback, which has greatly improved the quality of our work. We are happy to answer any further questions or follow any requests, your advice is crucial for improving the quality of our work.
>
> Sincerely,
>
> The Authors
>
> [1] Mallya, Arun and Lazebnik, Svetlana, "PackNet: Adding Multiple Tasks to a Single Network by Iterative Pruning”
>
> [2] Lyle, Clare, Mark Rowland, and Will Dabney. “Understanding and preventing capacity loss in reinforcement learning”
>
> [3] Lyle, Clare, et al. “Understanding plasticity in neural networks”
>
> [4] Stripelis, Dimitris, et al. "Federated progressive sparsification (purge, merge, tune)+.”
>
> [5] Wang, Yangyang, et al. "Theoretical convergence guaranteed resource-adaptive federated learning with mixed heterogeneity.”
>
> [6] Zhou, Hanhan, et al. "Every parameter matters: Ensuring the convergence of federated learning with dynamic heterogeneous models reduction."

---

> > ### Comment · Reviewer_Ay19 · 2024-11-26
> >
> > Thanks for the thoughtful response. My main concerns have been addressed and I have raised my score.

---

> > > ### Author Response · Authors · 2024-11-28
> > >
> > > Dear Reviewer Ay19,
> > >
> > > Thank you for your thoughtful response and for raising your score. We greatly appreciate your feedback and are glad to hear that your main concerns have been addressed.
> > >
> > > To further strengthen our work, we want to emphasize that we have implemented a **heterogeneous sparsity setting**, following the heterogeneous setting proposed in FLASH. In this setup, clients were divided into groups, each training with a different sparsity level, simulating heterogeneous hardware capabilities. SparsyFed demonstrated an impressive ability to adapt to heterogeneous sparsity levels among clients, consistently outperforming other baselines by a significant margin in the most extreme settings. Details about this additional experiment are provided in Section E.5 of the appendix.
> > >
> > > Please do not hesitate to reach out if you have additional questions or suggestions. We are happy to provide further clarifications or address any remaining concerns to improve the manuscript.
> > >
> > > Sincerely,
> > >
> > > The Authors

---

### Official Review · Reviewer_LGTa · 2024-11-03

**Soundness:** 3
**Presentation:** 2
**Contribution:** 2
**Rating:** 6
**Confidence:** 4

**Summary:**

SparsyFed introduces a novel approach to federated sparse training that achieves high model sparsity (95%) while maintaining accuracy. To create a practical federated sparse training method that addresses the issues of data heterogeneity, adaptability, and hyperparameter tuning, SparsyFed produces highly sparse models with negligible accuracy degradation, requires only one hyperparameter, achieves significant weight regrowth reduction, and maintains model plasticity under sparse design.

**Strengths:**

1. Only needs one hyperparameter (sparsity target), making it very practical for FL settings where tuning is challenging
2. Maintains accuracy up to 95% sparsity, significantly outperforming baselines
3. Good ablation studies on weight reparameterization and activation pruning

**Weaknesses:**

1. The explanation would be clearer if a figure illustrating the pipeline of the proposed method were provided.
2. This reviewer believes that focusing on fewer hyperparameter tuning efforts should not be considered the main contribution (or even the primary motivation), especially since addressing this challenge isn't the primary focus of the work. Furthermore, while the proposed method introduces only one hyperparameter, existing hyperparameter-related issues in FedOpt still persist.
3. The reviewer finds it confusing that Line 10 in Algorithm 2 appears redundant given the presence of Line 9.
4. Is sparse communication essential for the proposed method to address the three challenges it focuses on?
5. The paper lacks sufficient discussion on why the proposed method facilitates more effective consensus on the sparse pattern.
6. There is an absence of theoretical analysis or guarantees regarding the convergence properties of the method.

**Questions:**

see weaknesses.

---

> ### Author Response · Authors · 2024-11-27
>
> Dear Reviewer LGTa,
>
> Thank you for your detailed comments and for highlighting the practical benefits of SparsyFed. Your feedback has been extremely helpful, and we have carefully addressed your concerns as follows:
>
> 1. **Pipeline Clarity**
>     - To improve clarity, we have added a diagram illustrating the pipeline of our proposed method. It is a visual overview of the key steps of our training pipeline, with a particular focus on the sparse training pipeline on the client. To enhance the overall clarity of the manuscript we decided to put in the introduction (new Fig.1) to give a clear and immediate view of the work we are proposing.
> 2. **Algorithm Clarification**
>     - In Algorithm 2, we retained both Line 9 (layer-wise pruning of activations) and Line 10 (updating activation values globally) to highlight their distinct roles. Line 9 involves operations applied at the layer level, while Line 10 updates the entire model. This distinction ensures clarity and prevents misinterpretation. We have revised Section 3, particularly steps 6–10 of the algorithm, to improve the clarity of each step. Specific updates were made in the subsections *Sparsity-Inducing Weight Reparameterization* and *Pruning Activations During Local Training* to elaborate on the rationale behind these steps and their contribution to the overall design of the algorithm.
> 3. **Hyperparameter Tuning Efforts**
>     - We understand your concern that hyperparameter tuning should not be emphasized as the primary contribution. However, we believe that our focus on reducing hyperparameter complexity, while not resolving all related challenges in FedOpt, is an often-overlooked aspect in other works. This simplification ensures practicality and usability, particularly in federated learning scenarios. Our design avoids introducing additional hyperparameters, which can exacerbate tuning difficulties, and instead uses only a single hyperparameter (sparsity target) to balance performance and adaptability.
> 4. **Sparse Communication Relevance**
>     - While sparse communication is not the primary focus of our research, it is a critical consideration in federated learning due to communication being a major bottleneck in such systems. Reducing communication costs during training can significantly improve overall efficiency, as both the model and local updates must be transmitted between clients and the server in every round.
>     - SparsyFed achieves substantial reductions in both uplink and downlink communication, as illustrated in Fig. 2(a). This reduction translates into noticeable speedups of the overall training and lower bandwidth requirements, particularly beneficial for cross-device FL environments.
> 5. **Consensus on Sparsity Patterns**
>     - We have expanded the discussion in Section 5.3 to explain why SparsyFed facilitates better consensus on sparsity patterns across clients. This improvement came from the weight reparameterization technique, which enables the model to intrinsically focus training on a consistent subset of weights. This alignment enhances convergence and performance across heterogeneous client data distributions.
> 6. **Theoretical Guarantees**
>     - As this work focuses on practical implementations and experimental validation, we did not include a theoretical analysis of convergence properties. Such an analysis would be outside the scope of this paper, but we acknowledge its importance and consider it a potential direction for future work.
>
> We hope these responses address your concerns comprehensively. Thank you again for your constructive feedback, which has helped us refine our paper.
>
> Sincerely,
>
> The Authors

---

> > ### Comment · Reviewer_LGTa · 2024-11-28
> >
> > some of my concerns have been addressed. I would raise my score.

---

> > > ### Author Response · Authors · 2024-11-28
> > >
> > > Dear Reviewer LGTa,
> > >
> > > Thank you for your updated feedback and for raising your score. We appreciate your acknowledgement of the improvements made. If there are any remaining concerns or areas where we can provide further clarification please don’t hesitate to let us know, we will do our best to address them.
> > >
> > > Sincerely,
> > >
> > > The Authors

---

### Official Review · Reviewer_wQKr · 2024-11-03

**Soundness:** 3
**Presentation:** 3
**Contribution:** 3
**Rating:** 8
**Confidence:** 2

**Summary:**

SparsyFed is designed for cross-device environments where devices have limited computational and communication resources. Traditional federated learning approaches struggle with communication overhead, computational demands, and data heterogeneity across devices, especially with dense models. SparsyFed addresses these challenges by introducing sparse adaptive federated training, which reduces the model’s memory and computational requirements while retaining high accuracy. The key innovations include a sparsity-inducing weight reparametrization technique that requires only a single additional hyperparameter and a method for pruning activations to reduce computational load on clients. SparsyFed achieves up to 95% sparsity with minimal accuracy loss, which is particularly beneficial for constrained edge devices in cross-device FL.  SparsyFed demonstrates significant improvements over traditional and recent sparse FL baselines, achieving a 19.29× reduction in communication costs and faster global model convergence. The challenges of data heterogeneity, limited client resources, and the need for high sparsity without complex hyperparameter tuning are addressed through its adaptive and lightweight design.

**Strengths:**

This paper is well written and has a solid motivation. The strength of this paper is as follows,

1. SparsyFed can achieve target sparsity levels of up to 95% while maintaining competitive accuracy. This is particularly beneficial in federated learning environments, where models often face challenges from heterogeneous data distributions​.

2. The method significantly reduces communication overhead, achieving up to 19.29 times less communication than dense models.

3. This is crucial in cross-device federated learning where devices often have limited bandwidth unlike many federated learning methods that require multiple hyperparameters, SparsyFed simplifies the tuning process by needing only one hyperparameter.

**Weaknesses:**

This paper is experimentally solid but theoretically weak. The weaknesses of this paper are as follows,

1.	The authors mentioned the model’s sparsity by reducing the FLOPs and memory footprint in lines 207 and 208. However, the authors have not given any experimental evaluation of it. It would be good to give a FLOP analysis of SparsyFed with the dense model.

2. In experiments (line 325) “19.29×less communication costs compared to the dense model” what is the convergence bound of the proposed algorithm? Is it similar to FedAvg? Why is SparsyFed more efficient in communication than state-of-the-art?

3. I do not understand Figure 1 (right). Is it the comparison of the different local models or the global model? What happens when the target sparsity is more than 90%? I think author should elaborate more on  Consensus on the spare masks across clients (Section 5.3).

4. In Table 2, If we increase sparsity why SparsyFed with activation pruning perform worse in both IID and non-IID scenarios? What happens when $\alpha$ =0.1 ? The inference from the observation is not clear to me.

5. In Figure 3, the authors proposed these reparameterization methods or those available in the state-of-the-art ? What authors tried to show here is unclear to me could author please elaborate more on Figure 3 ?

**Questions:**

Asked in weakness

---

> ### Author Response · Authors · 2024-11-27
>
> Dear Reviewer wQKr,
>
> Thank you for your insightful feedback and for recognizing the strengths of our experimental evaluation. Your detailed critique has provided us with valuable guidance for improving our work. Below, we address your comments point by point:
>
> 1. **Clarifying Results and Captions**
>     - We clarified that Fig. 2 (right) compares the sparsity levels achieved by the global model after each round of training. These values are measured after aggregating local updates on the server. Due to the non-exact overlap of the local updates, the global model tends to gain density after aggregation. A smaller density gain indicates that the updates share most weights in their binary masks, reflecting a better consensus on the masks among clients.
>     - Additional plots with varying target sparsity levels have been added to the appendix to provide a broader perspective, fig. 6 in section E.3.
>     - Section 5.3.1 has been rewritten to better elaborate on the concept of mask consensus. Improved consensus results from the weight reparameterization technique, which encourages clients to focus on the same subset of weights. This leads to marked mask stability without excessive weight growth, resulting in convergence to a shared mask.
> 2. **Activation Pruning Performance**
>     - When comparing SparsyFed with and without activation pruning, performance degradation was noticeable only at extreme sparsity levels. In such cases, some layers tend to become extremely sparse to meet the overall target sparsity. Pruning the activations of these specific layers can significantly reduce the information flow between layers, thereby impacting overall performance. It is important to note that this phenomenon occurs only at very high sparsity levels. At lower, yet significant, sparsity levels, our approach shows negligible differences in performance.
>     - Section 5.5 and Table 2 have been updated to clarify this phenomenon. Experiments for **α = 0.1** have been included in Table 2 in the updated submission.
> 3. **Reparameterization Method**
>     - We selected reparameterization methods based on the intuition that an inherently sparse model would perform better when a layer-tailored sparsification of activations is applied. These methods were chosen for their ability to induce inherent sparsity during training while mitigating performance degradation caused by sparsity. Section 5.4 has been updated to clarify our rationale and explain how these methods were integrated into SparsyFed, as well as why we adopted the reparameterization proposed in Powerpropagation.
>     - Regarding Fig. 4, the reparameterization methods shown were selected from state-of-the-art approaches to evaluate their ability to induce sparsity naturally and their impact on performance under high sparsity levels.
> 4. **Theoretical Analysis of Costs**
>     - The computational reduction achieved through the use of sparse matrices during training and its associated benefits have been discussed in previous works like [1] in a centralized environment, and in [2] in a federated context. Our main contribution lies in recognizing the varying nature of the layers in a neural network, where different layers retain different amounts of information. Unlike previous approaches, which applied the same level of sparsity across all layers, we address this issue by using the layer-wise sparsity, as computed in step 8, instead of the target sparsity. This approach allows for a more tailored pruning strategy that aligns with the information retention characteristics of each layer.
> 5. **Convergence Analysis**
>     - While theoretical convergence bounds are beyond the current scope of the paper, we have clarified why SparsyFed demonstrates better performance compared to state-of-the-art methods. Specifically, as shown in Fig. 2(b), our method achieves high sparsity almost immediately, resulting in reduced downlink communication (server to client) after the initial round. Additionally, the weight reparameterization technique helps clients focus on a consistent subset of weights, leading to a faster convergency, as it’s possible to see in Fig. 2(a) SparsyFed reaches better accuracy results compared for example to Top-K, with an overall similar communication cost.
>
> We hope these responses address your concerns thoroughly. We greatly appreciate your constructive feedback and have used it to refine both the clarity and depth of our paper. Your critique has been instrumental in strengthening our work, and we look forward to any further suggestions you may have.
>
> Sincerely,
>
> The Authors
>
> [1] Raihan, Md Aamir, and Tor M Aamodt. "Sparse weight activation training." In Advances in Neural Information Processing Systems, December 2020.
>
> [2] Qiu, Xinchi, Javier Fernandez-Marques, Pedro PB Gusmao, Yan Gao, Titouan Parcollet, and Nicholas Donald Lane. "Zerofl: Efficient on-device training for federated learning with local sparsity." In International Conference on Learning Representations, 2021.

---

> > ### Comment · Reviewer_wQKr · 2024-11-28
> >
> > The authors have addressed all my concerns except one (FLOP analysis). Moreover, SpersyFed with activation pruning performs worse when the sparsity level is too high, regardless of IID or non-IID datasets. Therefore, I believe this problem is a generic sparse ML problem, not an FL-specific problem (Data heterogeneity). This is a limitation of the current work. However, I appreciate the authors' efforts. The paper is much clearer to me now. Therefore, I would like to increase my score.

---

> > > ### Author Response · Authors · 2024-11-28
> > >
> > > Dear Reviewer wQKr,
> > >
> > > Thank you for your thoughtful feedback and for increasing your score.
> > >
> > > Regarding the FLOP analysis, we understand its importance and are actively working on incorporating a tailored evaluation in future revisions. We welcome any specific suggestions you might have on this.
> > >
> > > We also acknowledge your point on SparsyFed's performance at extreme sparsity levels. As noted, this reflects a broader sparse ML challenge rather than an FL-specific issue. Still, we would like to point out that we managed to limit this degradation compared to the other baseline, like ZeroFL [1], that performs activation pruning with a fixed threshold for all the layers of the network.
> > >
> > > We appreciate your constructive input and are happy to address any further questions or suggestions you may have.
> > >
> > > Thank you again for your support and valuable insights.
> > >
> > > Sincerely,
> > >
> > > The Authors
> > >
> > > [1] Qiu, Xinchi, Javier Fernandez-Marques, Pedro PB Gusmao, Yan Gao, Titouan Parcollet, and Nicholas Donald Lane. "Zerofl: Efficient on-device training for federated learning with local sparsity." In International Conference on Learning Representations, 2021.

---

> > > > ### Comment · Reviewer_wQKr · 2024-11-29
> > > > **Flop analysis**
> > > >
> > > > Hi, You can use profiling tools like ptflops. This library computes theoretical FLOPs for PyTorch models. It provides per-layer computational costs and supports various architectures, including convolutional and transformer-based models.

---

> > > > > ### Author Response · Authors · 2024-12-03
> > > > >
> > > > > Dear Reviewer wQKr,
> > > > >
> > > > > We appreciate you pointing us to a potentially useful tool. However, ptflops is designed to compute the theoretical complexity of a model based solely on its structure, without considering or leveraging sparsity. This means that, while the tool can calculate the computational cost of the ResNet18 architecture (as used in our work), it cannot measure the difference in FLOPs between SparsyFed and the other baselines, as all are applied to the same underlying model.
> > > > >
> > > > > We strongly agree on the importance of using an appropriate tool to measure the FLOPs reduction metrics, and we will continue to work on identifying and integrating a solution that can effectively evaluate our use case.
> > > > >
> > > > > Thank you again for your support and valuable insights.
> > > > >
> > > > > Sincerely,
> > > > >
> > > > > The Authors

---

### Official Review · Reviewer_yVEF · 2024-11-03

**Soundness:** 3
**Presentation:** 3
**Contribution:** 2
**Rating:** 5
**Confidence:** 5

**Summary:**

This paper discusses utilizing sparse learning to reduce communication and computation costs in cross-device federated learning. The authors propose the SparsyFed algorithm, where in every round, the participants first reparameterize the weights and then perform the forward pass. In the next step, before calculating the gradient, each client prunes the activations in each layer. Finally, it only sends back the TopK large gradients to the server. This way, they can reduce client communication costs to the server.

**Strengths:**

* The writing, motivation, and related works have been clearly explained.

* The algorithm achieves high sparsity ratios and communication cost reduction with low-performance degradations.

* The authors have presented interesting results in the evaluation section.

**Weaknesses:**

* The novelty of the algorithm is not apparent.

* While the paper is well-written, and the authors have successfully conveyed the problem statement, the algorithm is not clearly explained. For example, how does step 8 work?

* It is not clear why different decisions have been made and what effect each of them has on the final performance. For example, why does weight reparameterization work, or how does layer-wise top-k selection result in layers with different sparsities?

* The experiment section only covers resent18, and it is unclear if this method works on other types or sizes of models.

**Questions:**

* Why did not the authors not consider other existing works they mention as their baseline?

* Could the authors explain the distinction between their paper and a naive federated version of [Schwarz et al. (2021)]?

* Is the sensitivity of this method to client participation rate, number of clients, and number of data points within each client? Is it possible that when some clients do not have enough data, SparsyFed might fail to converge or find a proper sparse model?

* What is the computation cost of steps 8 - 9 - 10 in algorithm 2 compared to full training?

* In the appendix, the authors presented two versions of the FLASH algorithm, one of which does not fix the mask (G.4). Could the authors explain why they selected G.3 over G4? Because it seems that G4 does not have the plasticity problem.

---

> ### Author Response · Authors · 2024-11-27
>
> Dear Reviewer yVEF,
>
> We sincerely thank you for your thoughtful feedback and for highlighting both the strengths and weaknesses of our work. Below, we address your comments and outline the improvements we have made or plan to implement based on your valuable input:
>
> 1. **Clarity of the Algorithm**
>     - We revisited the introduction to clarify the novelty and technical aspects of the proposed algorithm. We also added a graphical pipeline to show how our method works in new Fig. 1.
>     - We have revised Section 3, particularly steps 6–10 of the algorithm, to enhance the clarity of each step of the algorithm. Specific updates were made in the subsections to explain the reasoning behind the steps, and how they contribute to the overall design of the algorithm. We added the missing definitions of the SetLayer and GetLayer functions.
>     - Regarding Alg. 2, in step 8, we compute per-layer sparsity levels for each layer. Such per-layer sparsity levels reflect the per-layer sparsity of weights (the fraction of zero values in the weights of the specific layer). In step 9 of Alg.2, activations are pruned layer-wise via TopK operation using the per-layer sparsity level computed at step 8. This approach allows for a more tailored pruning strategy that aligns with the information retention characteristics of each layer.
>     - We clarify here that the model **weights** are pruned applying the target sparsity level to the entire neural network, leading to a non-uniform level of sparsity among the layers of the network.
> 2. **Weight Reparameterization and Sparsity**
>     - To clarify why we used a weight reparametrization method and what is the intuition beyond his effectiveness we updated the evaluation section. Specifically, in Section 5.4, we explain how the chosen weight reparameterization aims to train a sparse model helping the resilience of pruning. The consequences of this can be observed in the sparsity pattern of the model, which we detail in Section 5.3. Finally, in Section 5.5, we discuss the impact of activation pruning.
> 3. **Computation Reduction**
>     - The computational reduction achieved through the use of sparse matrices during training and its associated benefits have been discussed in previous works like [1] in a centralized environment, and in [2] in a federated context. One of our main contributions lies in the application of a per-layer sparsity to the activations. Previous approaches such as SWAT [1] apply a fixed sparsity level across the activations of all layers. In contrast, we use a layer-wise sparsity for activations (step 8 of Alg. 2), instead of the target sparsity. Such per-layer sparsity levels (applied to the activations) reflect the per-layer sparsity of weights (the fraction of zero values in the weights of the specific layer).
> 5. **Baseline Comparisons**
>     - We implemented a naive federated version of Powerpropagation as a baseline for comparison. In the original Powerpropagation paper, pruning was applied at the end of centralized training. We trained the model in a federated setting on various data heterogeneity partitions, then pruned and evaluated it at the end of the training. As evident from the reported results, SparsyFed significantly outperforms its naive counterpart. A detailed description of the naive federated Powerprogation will be provided in the appendix, SectionE.1.
>     - Regarding the selection of G.3 over G.4 from the appendix: The performance of the two FLASH variants (G.3 and G.4) were similar in the reported results of the original paper, with the former obtaining better performance in some settings. We decided to use G3 since it uses fewer hyperparameters, and, in this sense, it is more similar to our method, which only requires one sparsity-related hyperparameter.
> 6. **Sensitivity Analysis**
>     - We recognize the importance of understanding how client participation, number of clients, and data heterogeneity affect our method. We integrated in the appendix an experiment on CIFAR-10 with a large federation of clients, and a few local samples for each participant. We used a low level of participation rate of 0.1% which translates to 10 clients per round. Our results show how our method outperforms other baselines in this setting. A more detailed description, along with the results, has been included in the appendix in Section E.6.
>
> We hope these responses address your concerns thoroughly. Your feedback has been invaluable in guiding improvements to our work, and we are committed to ensuring our revisions meet the highest standards. Thank you again for your constructive critique.
>
> Sincerely,
>
> The Authors
>
> [1] Raihan, Md Aamir, and Tor M Aamodt. "Sparse weight activation training." In Advances in Neural Information Processing Systems, December 2020.
>
> [2] Qiu, Xinchi, Javier Fernandez-Marques, Pedro PB Gusmao, Yan Gao, Titouan Parcollet, and Nicholas Donald Lane. "Zerofl: Efficient on-device training for federated learning with local sparsity." In ICLR, 2021.

---

> > ### Comment · Reviewer_yVEF · 2024-11-29
> > **Response to Authors**
> >
> > I want to thank the authors for their explanations. However, some of my initial questions remain unresolved.
> >
> > * Based on my understanding, line 8 in Algorithm 2 only calculates the zeros/total num params ratio. Does it mean that there is no control over layer-wise sparsity? Can this method cause layer collapse or mostly uniform sparsities?
> >
> > * I still believe ResNet models are very resilient to layer collapse because of their structure. However, they are not the most commonly used model in a federated setting. The authors could show that their method is also applicable to other standard model families; otherwise, it limits the scope of the paper.
> >
> > * About the Powerpropagation baseline, is the only difference between Powerpropagation and Sparsyfed the fact that in Sparsyfed, the same pruning is applied at the **end of every FL round** instead of at the **end of the training** (besides the natural differences between FL and centralized training)?

---

> > > ### Author Response · Authors · 2024-12-02
> > >
> > > Dear Reviewer yVEF,
> > >
> > > Thank you for your feedback and for pointing out the remaining concerns. We appreciate the opportunity to clarify further:
> > >
> > > 1. **Layer-wise sparsity control**: You are correct that our method does not enforce uniform sparsity across layers. Pruning is applied globally across the model's weights, resulting in non-uniform layer-wise sparsity due to differences in layer dimensions and roles within the architecture. This approach was deliberate, as enforcing fixed sparsity across layers often degrades performance. For models more sensitive to layer collapse, alternative pruning techniques could replace the current method (e.g., in Line 9, Algorithm 1) to better control sparsity distribution and, consequently, activation pruning (Line 8, Algorithm 2).
> > > 2. **Model selection and scope**: We chose ResNet models and this particular setting to ensure a fair comparison with existing baselines in the federated learning field. It is in our interest to expand our assessment to other model families as a direction for future work, and we welcome suggestions on alternative models or settings that would enhance the paper's scope.
> > > 3. **Comparison with Powerpropagation**: While it is true that *SparsyFed* applies pruning and retraining at the end of each FL round, our contribution extends beyond this. Specifically, our work explores how the combination of reparameterization and pruning behave in an FL environment, from the raw accuracy performance to the influences to convergence of sparsity masks. We highlight that Powepropagation is a specific implementation of a general reparametrization method in the Sparsyfed pipeline. Although Powerpropagation reported the best performance in our experiments, alternative reparameterization methods were also explored and could be applied, as shown in our ablation study, section 5.4. Regarding the comparison between *SparsyFed* and naive Powerpropagation, while this simplification holds to some extent, it represents only a narrow aspect of the broader contributions of our work, which focuses on the synergy of its components in an FL setting.
> > >
> > > Please let us know if you have any additional suggestions or specific directions you would like us to explore. Your insights are highly valued.
> > >
> > > Sincerely,
> > >
> > > The Authors

---

### Meta-Review · Area_Chair_vrgq · 2024-12-18

**Metareview:**

Summary of Scientific Claims and Findings:
SparsyFed is a novel approach for cross-device federated learning that achieves high model sparsity (up to 95%) while maintaining accuracy. The key innovations include:

- A sparsity-inducing weight reparameterization technique requiring only one hyperparameter
- Layer-wise activation pruning to reduce computational load on clients
- An approach that addresses data heterogeneity and model plasticity challenges

Main Strengths:

- Strong experimental results showing significant communication cost reduction (19.29× less) while maintaining competitive accuracy
- Simple hyperparameter tuning requirements compared to existing approaches
- Comprehensive ablation studies validating the key components
- Clear practical benefits for resource-constrained federated learning environments

Main Weaknesses:

- Limited theoretical analysis and convergence guarantees
- Experiments focused mainly on ResNet architecture
- Some lack of clarity in algorithm presentation, particularly around steps 8-10 in Algorithm 2
- Missing FLOP analysis for computational efficiency claims

**Additional Comments On Reviewer Discussion:**

Outcomes from Author-Reviewer Discussion:
The authors have addressed many initial concerns through their responses and proposed revisions:

- Added visual pipeline diagram to clarify the method
- Expanded explanation of layer-wise sparsity approach
- Clarified the role of weight reparameterization in achieving consensus
- Acknowledged limitations regarding theoretical analysis
- Added experiments with heterogeneous sparsity settings

Reviewer Agreement/Disagreement:
Initial ratings ranged from 3 to 8, with most reviewers increasing their scores after author responses. Final consensus emerged around accepting the paper, with reviewers acknowledging the practical value despite some theoretical limitations.

Suggestions for Improvement:

- Add FLOP analysis using appropriate profiling tools
- Expand theoretical analysis of convergence properties
- Include experiments with additional model architectures
- Better explain the relationship between activation pruning and computational efficiency
- Clarify the mechanism of plasticity claims

---

### Decision · Program_Chairs · 2025-01-22

Accept (Poster)